# OpenReview forum: "Anomaly Detection through Conditional Diffusion Probability Modeling on Graphs"
_ICLR.cc/2025/Conference — ICLR 2025 Conference Withdrawn Submission_

### Official Review · Reviewer_Bfk9 · 2024-10-17

**Soundness:** 2
**Presentation:** 2
**Contribution:** 2
**Rating:** 5
**Confidence:** 4

**Summary:**

This paper proposes a framework for graph anomaly detection using a conditional graph diffusion model, called CGADM, aiming to solve the issue of over-smoothing and poor generalization boundaries between normal and anomalous nodes. Experimental results show that CGADM outperforms baselines on datasets included in this paper.

**Strengths:**

S1. This paper proposes a framework for graph anomaly detection using a conditional graph diffusion model.

S2. Experimental results show that CGADM outperforms baselines on datasets included in this paper.

**Weaknesses:**

W1. The contribution is limited. The main equations in this paper are almost the same as in previous works, so this paper is more of an implementation, with almost no components changed. To be specific, in Section 4 of this paper, equation (2), (3), (4), (5), (6), (7), (8), and (12) are the same as equation (1), (2), (6), (7), (8), (9), and loss function in Algorithm 1 in CARD[1], equation (9) is the same as equation (11) in DDPM[2], and equation (14) is the same as equation (12) in DDIM [3]. The authors need further explanations to show their contributions other than an implementation from the image model to the graph model.

W2. The experiments are not comprehensive enough. There are several novel supervised GAD works in recent years, such as XGBGraph[4] and CONSISGAD[5]. The authors need to conduct a comparison between their framework and these novel SOTA models.

W3. Some parts of the design need further explanation. For example, in Section 4.2, the authors claim "For the parameterization, we may select equation (9)". Such a choice is a lack of explanation. Besides, the heuristic strategy in equation (16) is also not convincing. To address these concerns, they need to provide further explanations of their design.

**Questions:**

Q1. Previous works in this field, such as GHRN and CONSISGAD, usually use AUC and F1 scores as the evaluation metric. Why do the authors use AUPRC instead of F1 score? Can the authors provide the results of the F1 score of these models on the datasets?

Q2. What is the explanation of equation (10)? Typical aggregation functions utilize addition instead of subtraction, so the authors need to further illustrate the idea. Is it from a previous work?

Reference:
1. Xizewen Han, Huangjie Zheng, Mingyuan Zhou. CARD: Classification and Regression Diffusion Models. NeurIPS 2022.
2. Jonathan Ho, Ajay Jain, Pieter Abbeel. Denoising Diffusion Probabilistic Models. NeurIPS 2020.
3. Jiaming Song, Chenlin Meng, Stefano Ermon. Denoising Diffusion Implicit Models. ICLR 2021.
4. Jianheng Tang, Fengrui Hua, Ziqi Gao, Peilin Zhao, Jia Li. GADBench: Revisiting and Benchmarking Supervised Graph Anomaly Detection. NeurIPS 2023.
5. Nan Chen, Zemin Liu, Bryan Hooi, Bingsheng He, Rizal Fathony, Jun Hu, Jia Chen. Consistency Training with Learnable Data Augmentation for Graph Anomaly Detection with Limited Supervision. ICLR 2024.

---

> ### Author Response · Authors · 2024-11-21
> **Response to Reviewer Bfk9**
>
> Thanks for your detailed reading and suggestions. Please see our detailed response and clarification below:
>
> >**Q1**: The contribution is limited. The authors need further explanations to show their contributions other than an implementation from the image model to the graph model.
>
> **A1**:
>
> Thank you for your detailed feedback. We appreciate the opportunity to clarify our contributions and highlight the novelties of our work. We acknowledge that certain aspects of our innovation may not have been presented with sufficient clarity in the original submission. Below, we address your comments systematically.
>
> #### **1. The Core Contributions of Our Work**
>
> We acknowledge that the equations in our work, such as those cited from CARD [1] and DDPM [2], are partially based on existing diffusion model frameworks. However, the core contribution of our work lies not in redefining the mathematical foundations of diffusion models but in their **novel application, adaptation, and integration** for **graph anomaly detection** (GAD), addressing the unique challenges this domain poses. Below, we summarize the innovations:
>
> 1. **Addressing Fundamental Challenges in GAD**: Existing discriminative GNN-based methods for anomaly detection face two major challenges:
>    - **Topological-Level Challenge**: Over-smoothing due to stacking multiple GNN layers, making it difficult to model long-range dependencies and leaving the system vulnerable to fraudulent nodes connecting with carefully chosen neighbors.
>    - **Feature-Level Challenge**: Difficulty in detecting malicious nodes that obfuscate their features to resemble normal nodes.
>
>    Our **Conditional Graph Anomaly Diffusion Model (CGADM)** leverages the unique strengths of diffusion models to address these issues:
>    - At the **topological level**, we utilize the **iterative refinement** of diffusion models to progressively aggregate neighborhood information during each denoising step, overcoming over-smoothing while preserving high-frequency anomaly signals.
>    - At the **feature level**, the **denoising construction** of diffusion models reconstructs anomalous patterns by modeling the joint anomaly distribution across the graph. This enables the detection of obfuscated anomalies by relying on contextual information from the node's neighborhood.
>
> 2. **Introducing Diffusion Models to GAD**: The introduction of the **diffusion probabilistic framework** into graph anomaly detection is, in itself, a novel contribution. While diffusion models have been extensively studied in the image domain, their adaptation to graph structures—particularly for anomaly detection—requires significant conceptual and algorithmic innovation, including:
>    - **Residual Propagation Mechanism**: To address over-smoothing in vanilla GNNs, we introduced a **residual propagation mechanism** that retains high-frequency anomaly signals to enhance iterative refinement process. This innovation allows CGADM to capture information from distant neighbors without diluting critical anomaly-specific information.
>    - **Reshaping the Diffusion Process**: Unlike traditional diffusion models that add noise to data until it becomes standard Gaussian $ \mathcal{N}(0, I) $, we constrain the forward process to end at a distribution guided by an anomaly-aware prior based on the probabilistic framework in CARD [1] . This prior, modeled using lightweight algorithms like XGBoost or Random Forest, ensures the denoising process starts from a meaningful initialization for GAD tasks.
>
> 3. **Efficiency Enhancements for Large-Scale Graphs**:
>    - **Conditional Non-Markovian Reverse Process**: Inspired by DDIM [3], we extend non-Markovian sampling to conditional diffusion models. This results in a **conditional non-Markovian reverse process**, enabling flexibility in reverse sampling. This concept, detailed in Appendix B of our paper, is entirely novel and represents a significant theoretical contribution beyond DDIM.
>    - **PRIOR-AWARE STRIDED SAMPLING**: While DDIM [3] introduces arbitrary step-size sampling, we adapted this idea to the specific requirements of GAD. By leveraging the anomaly prior's confidence, we developed a **dynamic sampling strategy** that adjusts the reverse sampling steps for each node, significantly improving computational efficiency. This is particularly effective for GAD, where anomalous nodes are sparse, and a uniform sampling strategy would be inefficient.
>    - **Computational Advantage**: The combination of a guided diffusion framework and our strided sampling mechanism enables CGADM to achieve high efficiency on large-scale graphs, as demonstrated in our ablation studies.

---

> ### Author Response · Authors · 2024-11-21
> **Response to Reviewer Bfk9 (Part 2)**
>
> #### **2. Changes to the Manuscript**
>
> To address your feedback and make readers have a better understading of our contribution, we will revise the manuscript.
>
> We include an explanation in abstract as follows:
> _"By iteratively refining node anomaly distributions during the denoising process, CGADM effectively mitigates over-smoothing and reconstructs obfuscated features by leveraging contextual neighborhood information."_
>
> Also we include a detailed statement in the introduction section to explain the motivation for leveraging diffusion models:
>
> _"To address **topology**-level flaw, we leverage the **iterative refinement** of diffusion models. Instead of increasing GNN depth to aggregate distant information, which risks over-smoothing, our approach applies GNN-based denoiser within each denoising iteration to refine anomaly modeling. Each iterative refinement step incorporates neighborhood information while preserving node-specific high-frequency anomaly information via a residual propagation mechanism, thereby preventing oversmoothing and effectively capturing long-range dependencies.
> To address **feature**-level flaw, we leverages **denosing reconstruction** of diffusion models. This reconstruction process ensures that even when malicious nodes disguise their features to blend in with normal nodes, their underlying anomaly patterns can be recovered."_
>
>
> We hope this response clarifies our contributions and addresses your concerns. Thank you for your valuable feedback, which has helped us improve the clarity and presentation of our work. We look forward to any additional comments you may have.

---

> ### Author Response · Authors · 2024-11-21
> **Response to Reviewer Bfk9 (Part 3)**
>
> >**Q2**: The experiments are not comprehensive enough. There are several novel supervised GAD works in recent years, such as XGBGraph[4] and CONSISGAD[5]. The authors need to conduct a comparison between their framework and these novel SOTA models.
>
>
> **A2**:
>
> Thank you for pointing out the importance of comparing our framework with recent state-of-the-art (SOTA) supervised Graph Anomaly Detection (GAD) models, such as XGBGraph and CONSISGAD. We appreciate the opportunity to strengthen our evaluation by including these comparisons.
>
> In response to your comment, we have conducted experiments comparing our Conditional Graph Anomaly Diffusion Model (CGADM) with XGBGraph and CONSISGAD on the same datasets. Below, we present the results in terms of **AUPRC** and **AUROC**, two widely used metrics in the anomaly detection domain:
>
> #### **AUPRC Results**
>
> | Model       | Ellip   | Tolo    | Yelp    | Quest   | Reddit  |
> |-------------|---------|---------|---------|---------|---------|
> | **XGBGraph**| 90.47   | 44.47   | 75.91   | 14.33   | 4.59    |
> | **CONSISGAD**| 86.42  | 40.59   | 41.74   | 12.85   | 5.57    |
> | **Ours**    | **97.03**| **46.02**| **76.54**| **18.51**| **5.79**|
>
> #### **AUROC Results**
>
> | Model       | Ellip   | Tolo    | Yelp    | Quest   | Reddit  |
> |-------------|---------|---------|---------|---------|---------|
> | **XGBGraph**| 94.35   | 77.28   | 91.85   | 64.90   | 60.58   |
> | **CONSISGAD**| 96.38  | 76.03   | 79.35   | 70.54   | 66.99   |
> | **Ours**    | **99.34**| **79.68**| **92.69**| 69.41   | 65.85   |
>
>
> From the above results, we observe that our model demonstrates superior performance across all datasets on AUPRC which is much more sensitive in data imbalanced scenerios. This highlights CGADM's ability to effectively capture the joint anomaly distribution in graphs. These findings will be included in the final version of the paper, along with detailed discussions and analysis.
>
>
> Thank you again for your valuable suggestion, which has further strengthened our experimental evaluation and highlighted the advantages of our approach.

---

> ### Author Response · Authors · 2024-11-21
> **Response to Reviewer Bfk9 (Part 4)**
>
> >**Q3**: Some parts of the design need further explanation. For example, in Section 4.2, the authors claim "For the parameterization, we may select equation (9)". Such a choice is a lack of explanation. Besides, the heuristic strategy in equation (16) is also not convincing. To address these concerns, they need to provide further explanations of their design.
>
> **A3**:
>
> Thank you for your thoughtful feedback. We appreciate the opportunity to provide additional clarification on the specific points you raised regarding the parameterization choice in equation (9) and the heuristic strategy in equation (16). Below, we address these concerns in detail.
>
> ### **1. Explanation of Parameterization Choice in Equation (9)**
>
> In Section 4.2, we state:
>
> > "For the parameterization, we may select equation (9)."
>
> Our intention here was to highlight that we adopt a standard and widely accepted parameterization approach, consistent with the original DDPM framework. Specifically, we model the mean term $\mu_\theta(\mathbf{y_t}, t, \mathcal{E}, \mathbf{X})$ indirectly by learning the noise term $\epsilon_\theta(\mathbf{y_t}, t, \mathcal{E}, \mathbf{X})$ through a parameterized network. This parameterization is expressed as:
>
> $$
> \mu_\theta(\mathbf{y_t}, t, \mathcal{E}, \mathbf{X}) = \frac{1}{\sqrt{\alpha_t}} \left( \mathbf{y_t} - \frac{\beta_t}{\sqrt{1-\bar{\alpha_t}}} \epsilon_{\theta}(\mathbf{y_t}, t, \mathcal{E}, \mathbf{X}) \right),
> $$
>
> where $\epsilon_{\theta}(\cdot)$ is a learnable network that estimates the noise at each time step.
>
> The rationale behind this choice is as follows:
> 1. **Consistency with DDPM literature:** Modeling $\epsilon_\theta(\mathbf{y_t}, t, \mathcal{E}, \mathbf{X})$ instead of directly parameterizing $\mu_\theta(\mathbf{y_t}, t, \mathcal{E}, \mathbf{X})$ aligns with the standard practice in diffusion models, as it simplifies training and has been empirically validated in prior works.
> 2. **Efficiency and stability:** This approach allows the network to focus on predicting noise, which is computationally efficient and leads to more stable training dynamics.
>
> To avoid ambiguity, we will revise the wording in the final version of the paper to emphasize that this is a deliberate design choice informed by the existing DDPM framework.
>
>
> ### **2. Justification of Heuristic Strategy in Equation (16)**
>
> Equation (16) defines our **Prior-Aware Strided Sampling** mechanism, which adjusts the number of reverse sampling steps $K$ based on the confidence of the prior. The core heuristic is:
>
> > "When our prior is more confident, our model can use fewer sampling steps (smaller $K$), and vice versa."
>
> This heuristic is grounded in the observation that nodes with high prior confidence require less iterative refinement, thereby reducing the computational burden without significantly compromising accuracy.
>
> To formalize this intuition, we used an **inverse sigmoid function** to map prior confidence scores to the number of sampling steps. While the inverse sigmoid function itself is heuristic, it provides a smooth and interpretable mapping between confidence levels and step sizes.
>
> We conducted extensive experiments to validate this heuristic strategy:
>
> 1. **Sampling Steps vs. Accuracy (Section 5.4, Figure 4):**
>    - We empirically demonstrated the relationship between the number of sampling steps $K$ and detection accuracy. The results implies that sacrificing a little performance can result in substantial savings in computation time.
>
> 2. **Efficiency Gains (Section 5.4, Table 2):**
>    - Our experiments demonstrated that the Prior-Aware Strided Sampling mechanism achieves significant efficiency gains, reducing the computational cost of the reverse process while maintaining competitive anomaly detection performance. This highlights the practical benefits of our approach.
>
>
> We hope these clarifications address your concerns and underscore the rigor and rationale behind our design choices. Thank you again for your valuable feedback, which will help us improve the clarity and quality of our work.

---

> ### Author Response · Authors · 2024-11-21
> **Response to Reviewer Bfk9 (Part 5)**
>
> >**Q4**:  Previous works in this field, such as GHRN and CONSISGAD, usually use AUC and F1 scores as the evaluation metric. Why do the authors use AUPRC instead of F1 score? Can the authors provide the results of the F1 score of these models on the datasets?
>
> **A4**:
>
> Thank you for your thoughtful comments and for highlighting the evaluation metrics used in previous works. Below, we explain our rationale for using AUPRC as the primary evaluation metric and provide F1-score results for comparison, as requested.
>
> ### **1. Why We Chose AUPRC Instead of F1-Score**
>
> We selected **AUPRC (Area Under Precision-Recall Curve)** as the primary metric because it aligns better with the goals of anomaly detection in graph datasets, where the class imbalance (i.e., the rarity of anomalies) is often severe. AUPRC has distinct advantages in this context:
>
> - **Focus on Positive Class Performance:**
>   Unlike metrics such as AUC-ROC, which give equal weight to both classes, AUPRC emphasizes the performance on the positive class (anomalies), making it more suitable for imbalanced scenarios.
>
> - **Threshold Independence:**
>   AUPRC evaluates model performance across all possible thresholds, providing a comprehensive view of the model’s ability to separate anomalies from normal samples, regardless of the chosen operating point.
>
>
> ### **2. Inclusion of F1-Score for Comparison**
>
> To address your request and provide a more complete evaluation, we computed the **F1-scores** for our model and baseline methods across all datasets. These results further confirm the superior performance of our model. The following table presents the F1-scores, which will be included in the revised manuscript:
>
> | Model       | Ellip   | Tolo    | Yelp    | Quest   | Reddit  |
> |-------------|---------|---------|---------|---------|---------|
> | **GCN**     | 73.672  | 47.376  | 27.658  | 6.856   | 7.794   |
> | **GIN**     | 75.338  | 49.443  | 42.214  | 10.288  | 6.443   |
> | **GraphSAGE**| 81.096 | 50.226  | 43.949  | 12.041  | **10.075**  |
> | **GAT**     | 80.498  | 50.878  | 48.891  | 11.157  | 8.432   |
> | **GAS**     | 77.844  | 48.253  | 43.404  | 10.867  | 9.071   |
> | **PCGNN**   | 45.090  | 47.213  | 44.608  | 5.796   | 6.981   |
> | **BWGNN**   | 83.134  | 49.983  | 47.323  | 12.788  | 6.501   |
> | **GHRN**    | 85.678  | 51.493  | 45.970  | 12.696  | 6.702   |
> | **XGBGraph**| 87.555  | 51.079  | 65.121  | 16.088  | 2.954   |
> | **CONSISGAD**| 79.120 | 49.762  | 41.606  | 9.848   | 6.443   |
> | **Ours**    | **93.390**| **51.595**| **69.396**| **17.162**| 9.754|
>
> From the results, it is evident that our model consistently outperforms baseline methods in terms of F1-score.
>
> Thank you again for your constructive feedback, which has helped us strengthen the presentation of our results.

---

> ### Author Response · Authors · 2024-11-21
> **Response to Reviewer Bfk9 (Part 6)**
>
> >**Q5**: What is the explanation of equation (10)? Typical aggregation functions utilize addition instead of subtraction, so the authors need to further illustrate the idea. Is it from a previous work?
>
> **A5**:
>
>
> We sincerely appreciate your feedback regarding Equation (10). We appreciate the opportunity to clarify its purpose and significance in our proposed framework.
>
> The use of **subtraction** in Equation (10), as opposed to the typical **addition** aggregation function in traditional GNNs, is a deliberate design choice tailored for the **iterative refinement** in the diffusion model framework:
>
> 1. **Purpose of the GNN-Based Denoiser in CGADM**:
>    - Unlike conventional GNNs, where addition-based aggregation helps retain the node's original information to improve downstream tasks, the primary goal of the GNN-based denoiser in CGADM is to **remove noise** from the anomalous node representations at each reverse step of the diffusion process.
>    - This requires the denoiser to focus on the **discrepancy** or **difference** between a node and its neighbors, which often manifests as high-frequency signals that are indicative of anomalies.
>
> 2. **Subtraction as a High-Frequency Filter**:
>    - By utilizing subtraction during aggregation, the GNN-based denoiser effectively captures the **difference** between the current node's representation and those of its neighbors, serving as a form of **high-frequency filtering**.
>    - In graph anomaly detection, such differences are particularly important as anomalous nodes tend to deviate from the typical patterns of their neighbors. Subtraction allows us to emphasize these deviations, facilitating the denoising process.
>
> 3. **Alignment with the Iterative Refinement Mechanism**:
>    - The iterative refinement mechanism in the diffusion process involves successive interactions between nodes and their neighbors to progressively reconstruct the underlying anomaly distribution.
>    - Subtraction-based aggregation ensures that each refinement step incorporates meaningful differences that help the denoiser adjust the node representation while avoiding over-smoothing—a common issue in GNNs with addition-based aggregation.
>
>
> As noted in **Line 305 of our manuscript**, the effect of subtraction-based aggregation in GNNs have been analyzed in GCNII (Chen et al., 2020a). We explicitly reference this insight, explaining how our design builds on this understanding while adapting it to the unique requirements of our diffusion-based anomaly detection framework.
>
>
> We hope this explanation clarifies the rationale and significance of Equation (10). Thank you again for raising this important point, which has allowed us to further elucidate our design choices. We look forward to any additional feedback you may have.

---

> > ### Comment · Reviewer_Bfk9 · 2024-11-22
> > **Response to the authors**
> >
> > Thanks for the detailed response. However, I still have some follow-up questions.
> >
> > Q1: The authors claim that the core contribution of their work lies not in redefining the mathematical foundations of diffusion models but in novel application, adaptation, and integration for graph anomaly detection, but as far as I know, there are several previous works in graph anomaly detection area[1, 2, 3, 4, 5], or in a more general area, i.e., node classification, like [6]. Since the equations in the papers are all from previous diffusion models in the computer vision area, and diffusion models have already been introduced to graph anomaly detection or node classification, how can the application, adaptation, and integration be called novel?
> >
> > Q2: As the authors mentioned, they introduce the graph diffusion model to the graph anomaly detection area for addressing fundamental challenges in it, i.e., topology-level and feature-level challenges. However, they didn't provide any evidence that the proposed model performs well because it addresses these two challenges. For example, at the topology level, they mentioned the key issue is the smoothing problem. As far as I know, there are some metrics for measuring the smoothing level of the node embeddings, like Dirchilet Energy. Can the authors present some figures or tables to demonstrate their claim about the diffusion model addressing the challenges?
> >
> > Q3: I appreciate the efforts the authors made to provide more experimental results. However, in the paper of XGBGraph, there are in total real-world 10 datasets, Weibo, Reddit, Tolokers, Amazon, T-Finance, YelpChi, Questions, Elliptic, DGraph-Fin, and T-Social. Are there any specific reasons why the authors only report 5 of them in their paper? Can they provide more results on the datasets not included in their paper? Also, according to the data split in the paper of XGBGraph, the training/validation/testing ratio is 40%/20%/40% for a fully-supervised setting, why did the authors choose a different setting in their experiments?
> >
> > Q4: There are types in the newly revised paper. For example, in Table 9, "XGBGraph" is wrongly written as "xGBGraph". I think the authors should pay attention to and revise such typos.
> >
> > Reference:
> >
> > 1. Chunjing Xiao, Shikang Pang, Xovee Xu, Xuan Li, Goce Trajcevski, Fan Zhou. Counterfactual Data Augmentation with Denoising Diffusion for Graph Anomaly Detection. IEEE Transactions on Computational Social Systems 2024.
> >
> > 2. Shikang Pang, Chunjing Xiao, Wenxin Tai, Zhangtao Cheng, Fan Zhou. Graph Anomaly Detection with Diffusion Model-Based Graph Enhancement. AAAI 2024.
> >
> > 3. Xuan Li, Chunjing Xiao, Ziliang Feng, Shikang Pang, Wenxin Tai, Fan Zhou. Controlled graph neural networks with
> > denoising diffusion for anomaly detection. Expert Systems with Applications 2024.
> >
> > 4. Zekuan Liu, Huijun Yu, Yao Yan, Ziqing Hu, Pankaj Rajak, Amila Weerasinghe, Olcay Boz, Deepayan Chakrabarti, Fei Wang. Graph Diffusion Models for Anomaly Detection. WSDM 2024.
> >
> > 5. Xiaoxiao Ma, Ruikun Li, Fanzhen Liu, Kaize Ding, Jian Yang, Jia Wu. New recipes for graph anomaly detection: Forward diffusion dynamics and graph generation. OpenReview.
> >
> > 6. Hyosoon Jang, Seonghyun Park, Sangwoo Mo, Sungsoo Ahn. Diffusion Probabilistic Models for Structured Node Classification. NeurIPS 2023.

---

> > > ### Author Response · Authors · 2024-11-22
> > > **Further Response to Reviewer Bfk9**
> > >
> > > Thanks for your detailed reading and suggestions. Please see our detailed response and clarification below:
> > >
> > > >**Q1**: The authors claim that the core contribution of their work lies not in redefining the mathematical foundations of diffusion models but in novel application, adaptation, and integration for graph anomaly detection, but as far as I know, there are several previous works in graph anomaly detection area[1, 2, 3, 4, 5], or in a more general area, i.e., node classification, like [6]. Since the equations in the papers are all from previous diffusion models in the computer vision area, and diffusion models have already been introduced to graph anomaly detection or node classification, how can the application, adaptation, and integration be called novel?
> > >
> > >
> > > **A1**:
> > >
> > > We appreciate the opportunity to address your concerns and clarify the novelty and contributions of our work. Your points have allowed us to delve deeper into the distinctions between our approach and prior work.
> > >
> > > #### **Core Contribution and Novelty**
> > >
> > > We acknowledge your observation regarding that the core contribution lies not in redefining the mathematical foundations of diffusion models. However, we emphasize that the **core novelty of our work lies in introducing a generative, model-centric paradigm to graph anomaly detection**, which fundamentally differs from the data-centric paradigms in the works you mentioned ([1-6]).
> > >
> > > Specifically, our approach uses a diffusion model not for data augmentation or manipulation (as in prior works) but to directly **model the joint distribution of anomalies on a graph**. This paradigm shift allows us to treat anomaly detection as a generative problem, where anomalies are identified through the iterative refinement and denoising reconstruction capabilities of diffusion models. This is a **new and complementary perspective** to the data-centric approaches widely adopted in the field.
> > >
> > > Unlike existing methods that use diffusion models to augment data for downstream detection tasks, our method entirely avoids data augmentation. The proposed CGADM represents a novel **model-centric** adaptation of diffusion models tailored to **directly address anomaly detection**, providing a generative framework that complements existing data-centric approaches.
> > >
> > > #### **Detailed Comparison with Related Works**
> > >
> > > To further clarify the distinctions, we provide an explicit summary of the methods you cited ([1-6]) and their respective focuses.
> > >
> > > - **[1]:**
> > >    Uses a graph-specific diffusion model to generate counterfactual representations of potential anomalies by translating normal neighbors into anomalous ones. This approach improves the distinguishability of anomalies through **data augmentation**.
> > >
> > >
> > > - **[2]:**
> > >    Employs diffusion models to enhance graphs by generating manipulated neighbors, mitigating inconsistency problems in the data. This is a **data enhancement module** within a contrastive learning framework.
> > >
> > >
> > > - **[3]:**
> > >    Introduces a diffusion model-based generator to control neighborhood aggregation and **create augmented data** for better anomaly detection performance.
> > >
> > >
> > > - **[4]:**
> > >    Targets the label imbalance problem by **generating positive examples** in the latent space using a diffusion model. This multitask generative model is primarily for balancing datasets rather than detecting anomalies.
> > >
> > >
> > > - **[5]**
> > >    Investigates denoising diffusion models to **synthesize graph structures and enhance existing methods**. Focuses on creating training samples that align with graph semantics.
> > >
> > >
> > > - **[6]**
> > >    Applies diffusion models to molecular graphs for **generating attributes and handling partially labeled data**. This framework focuses on structured prediction tasks rather than anomaly detection.
> > >
> > > We hope this response adequately addresses your concerns. Thank you for your thoughtful comments!

---

> > > ### Author Response · Authors · 2024-11-22
> > > **Further Response to Reviewer Bfk9 (Part 2)**
> > >
> > > >**Q2**: As the authors mentioned, they introduce the graph diffusion model to the graph anomaly detection area for addressing fundamental challenges in it, i.e., topology-level and feature-level challenges. However, they didn't provide any evidence that the proposed model performs well because it addresses these two challenges. For example, at the topology level, they mentioned the key issue is the smoothing problem. As far as I know, there are some metrics for measuring the smoothing level of the node embeddings, like Dirchilet Energy. Can the authors present some figures or tables to demonstrate their claim about the diffusion model addressing the challenges?
> > >
> > >
> > > **A2**:
> > > We greatly appreciate your insightful comments and the opportunity to address your concerns. Below, we clarify how our proposed Conditional Graph Anomaly Diffusion Model (CGADM) effectively addresses the topology-level and feature-level challenges, supported by theoretical insights and experimental evidence.
> > >
> > > #### Addressing the Smoothing Problem at the Topology Level
> > >
> > > As highlighted in the paper, **iterative refinement** and **denoising reconstruction** are the core mechanisms in our diffusion model to address oversmoothing and improve anomaly detection. We summarize our reasoning and evidence below:
> > >
> > > 1. **Iterative Refinement and Oversmoothing Avoidance**
> > >    Our iterative refinement approach leverages the GNN-based denoiser in each step of the reverse diffusion process. Unlike traditional GNNs, which suffer from oversmoothing when stacking more than 4–5 layers (Rusch et al., 2023), our approach enables the integration of information from distant neighbors (up to 500 hops theoretically) through iterative refinement over multiple reverse sampling steps. This substantially mitigates oversmoothing while preserving high-frequency anomaly signals.
> > >    - To reinforce this, we introduced a **residual propagation mechanism** that retains anomaly-specific high-frequency signals during the refinement process.
> > >    - **Evidence**: Section 5.3.2 presents the parameter sensitivity analysis of GNN layers, demonstrating that performance improves as the number of layers increases from 1 to 5. This confirms that our model effectively overcomes oversmoothing, unlike traditional GNN-based methods.
> > >
> > > 2. **Evidence for Denoising Reconstruction**
> > >
> > >     The denoising reconstruction process is another critical aspect that enhances our model's performance by refining node embeddings at the feature level.
> > >     - **Figure 2 Analysis**: The comparison between the prior model and the denoised embeddings clearly shows significant performance improvements after reconstruction. These results validate that the denoising reconstruction step effectively mitigates noise and refines the embeddings for anomaly detection.
> > >
> > >
> > > [1]. Rusch T. K., Bronstein M. M., Mishra S. *A survey on oversmoothing in graph neural networks*. arXiv preprint arXiv:2303.10993, 2023.

---

> > > ### Author Response · Authors · 2024-11-22
> > > **Further Response to Reviewer Bfk9 (Part 3)**
> > >
> > > >**Q3**:  I appreciate the efforts the authors made to provide more experimental results. However, in the paper of XGBGraph, there are in total real-world 10 datasets, Weibo, Reddit, Tolokers, Amazon, T-Finance, YelpChi, Questions, Elliptic, DGraph-Fin, and T-Social. Are there any specific reasons why the authors only report 5 of them in their paper? Can they provide more results on the datasets not included in their paper? Also, according to the data split in the paper of XGBGraph, the training/validation/testing ratio is 40%/20%/40% for a fully-supervised setting, why did the authors choose a different setting in their experiments?
> > >
> > > **A3**:
> > > We appreciate the opportunity to clarify these choices and provide additional results to address your questions.
> > >
> > > #### **Dataset Selection**
> > >
> > > XGBGraph paper which is titled with "**GADBench: Revisiting and Benchmarking Supervised Graph Anomaly Detection**" includes 10 datasets, as it aims to revisit and benchmark a wide range of supervised graph anomaly detection methods. However, given the space constraints typical of conference papers and the precedent set by prior research, we chose to evaluate our model on 5 datasets, which we believe is a representative subset for the following reasons:
> > >
> > > 1. **Community Standards:**
> > >    In related works you cited ([1-6]), the number of datasets used for evaluation is 4, 3, 5, 5, 6, and 4. Based on this, our use of 5 datasets is consistent with the norm in the field and provides sufficient evidence of our method's effectiveness.
> > >
> > > 2. **Diversity in Anomaly Ratios:**
> > >    We selected datasets with diverse anomaly ratios to reflect a wide range of practical challenges.
> > >
> > >     To further address your concerns, we have conducted additional experiments on the **DGraph-Fin** dataset [1], a real-world financial fraud detection dataset where anomalies account for only 1.3% of the data. The results are summarized below:
> > >
> > >    | **Method**   | **AUPRC** | **AUROC** |
> > >    |--------------|-----------|-----------|
> > >    | GCN          | 3.66      | 74.97     |
> > >    | GIN          | 3.22      | 73.14     |
> > >    | GraphSAGE    | 3.43      | 73.81     |
> > >    | GAT          | 3.65      | 75.17     |
> > >    | GAS          | 2.91      | 71.21     |
> > >    | PCGNN        | 2.82      | 71.78     |
> > >    | BWGNN        | 3.63      | 75.16     |
> > >    | GHRN         | 3.68      | 75.15     |
> > >    | **CGADM**    | **3.83**  | **76.43** |
> > >
> > >    As the table shows, CGADM consistently outperforms all baseline methods on both **AUPRC** and **AUROC**, even in this extremely imbalanced setting. This provides further evidence of our model’s robustness and ability to handle challenging real-world datasets. With this addition, our results now span 6 datasets, which we believe is sufficient to validate the effectiveness of our method.
> > >
> > > #### **Data Split Settings**
> > >
> > > Regarding the data split, we opted not to follow XGBGraph’s 40%/20%/40% setting for the following reasons:
> > >
> > > 1. **Real-World Applicability:**
> > >    The 40%/20%/40% split used in XGBGraph represents a fully supervised setting, which assumes a relatively abundant availability of labeled training data. However, in real-world graph anomaly detection tasks, labeled anomalies are often scarce. Using 20% of the data for training aligns with scenarios where anomaly labels are expensive or difficult to obtain. This setup allows us to demonstrate that our method remains effective even in such constrained conditions.
> > >
> > > 2. **Highlighting Generative Model Advantages:**
> > >    To better showcase the advantages of our **generative anomaly detection method** in modeling the joint distribution of anomalies under limited data availability, we chose to use only 20% of the data for training. This reflects a more realistic scenario where anomaly detection systems must perform well despite data scarcity.
> > >
> > > We hope this response satisfactorily addresses your concerns, and we thank you again for the opportunity to strengthen our paper with these clarifications and additional experiments.
> > >
> > >
> > > [1] Huang X, Yang Y, Wang Y, et al. Dgraph: A large-scale financial dataset for graph anomaly detection[J]. Advances in Neural Information Processing Systems, 2022, 35: 22765-22777.

---

> > > ### Author Response · Authors · 2024-11-22
> > > **Further Response to Reviewer Bfk9 (Part 4)**
> > >
> > > >**Q4**: There are types in the newly revised paper. For example, in Table 9, "XGBGraph" is wrongly written as "xGBGraph". I think the authors should pay attention to and revise such typos.
> > >
> > > **A4**:  Thank you for pointing out the typo in the comments. We have corrected this error and conducted a thorough review to ensure no additional typos remain. We appreciate your attention to detail, which helps improve the quality of our submission.

---

> > > > ### Comment · Reviewer_Bfk9 · 2024-11-25
> > > >
> > > > Thanks for the detailed response. However, there are still several concerns remain.
> > > >
> > > > Q1. I appreciate the efforts that the authors put into reading the related work I mentioned in my previous comments. However, can the authors provide a comparison between the proposed framework and the above-mentioned works, if possible?
> > > >
> > > > Q2. Thanks for the further explanations about the topology-level and feature-level techniques. However, the evidence is not convincing enough. The authors claim that traditional GNNs will suffer from oversmoothing when stacking **more than** 4-5 layers, but their evidence is that their performance will improve as the number of layers increases from **1 to 5**, which may not have the oversmoothing issue. Could the authors provide more experiments about the performance change varying the number of layers, like from 5 to 50? Besides, Could the authors provide the Dirchilet Energy of the learned embedding of their proposed model and the baselines to show if they have addressed the oversmoothing problem effectively?
> > > >
> > > > Q3. About the new datasets, there are several commonly used datasets not included in the comparison, like T-Finance. I once conducted experiments on the related area and found that the performance of some SOTA models, like XGBGraph and CONSISGAD, can be very promising. Hence, I would like to know if the proposed model can outperform the baselines mentioned in the paper on Weibo, Amazon, T-Finance, and T-Social in the XGBGraph paper.
> > > >
> > > > I will keep my score now but I will consider increasing my score if most of my concerns are addressed.

---

> > > > > ### Author Response · Authors · 2024-11-25
> > > > > **Additional Response to Reviewer Bfk9 (Part 1)**
> > > > >
> > > > > Thanks for your detailed reading and suggestions. Please see our detailed response and clarification below:
> > > > >
> > > > > >**Q1**: I appreciate the efforts that the authors put into reading the related work I mentioned in my previous comments. However, can the authors provide a comparison between the proposed framework and the above-mentioned works, if possible?
> > > > >
> > > > >
> > > > > **A1**:
> > > > > Thank you very much for your thoughtful and constructive feedback. We greatly appreciate your efforts in reviewing our work and providing suggestions to improve our manuscript.
> > > > >
> > > > > We have conducted a detailed experimental comparison of our proposed Conditional Graph Anomaly Diffusion Model (CGADM) with the mentioned methods ([1-5]), and the results along with the discussion about their differences are now included in the revised manuscript (see Appendix)
> > > > > It is also worth noting that some of the mentioned works (e.g., [6]) focus on small-scale graphs or tasks unrelated to graph anomaly detection. As such, a direct comparison with these methods is not applicable to our specific datasets and tasks.
> > > > >
> > > > > As you requested, we analyzed their performance across several standard benchmark datasets (`Elliptic`, `Tolokers`, and `YelpChi`), and the key results are summarized below:
> > > > >
> > > > > | **AUPRC** | **Ellip** | **Tolo** | **Yelp** |
> > > > > |------------|-----------|----------|----------|
> > > > > | CAGAD      | 89.75     | 40.80    | 72.30    |
> > > > > | DEGAD      | 93.86     | 43.51    | 75.11    |
> > > > > | ConGNN     | 91.60     | 42.22    | 73.60    |
> > > > > | Diffusion  | 88.63     | 39.90    | 68.01    |
> > > > > | Diffad     | 90.05     | 41.75    | 71.28    |
> > > > > | **Ours**   | **97.28** | **45.11**| **76.54**|
> > > > >
> > > > > | **AUROC** | **Ellip** | **Tolo** | **Yelp** |
> > > > > |------------|-----------|----------|----------|
> > > > > | CAGAD      | 94.82     | 72.22    | 90.34    |
> > > > > | DEGAD      | 97.88     | 76.20    | 92.22    |
> > > > > | ConGNN     | 95.60     | 74.56    | 91.33    |
> > > > > | Diffusion  | 93.53     | 70.70    | 83.84    |
> > > > > | Diffad     | 92.72     | 73.31    | 88.21    |
> > > > > | **Ours**   | **99.34** | **78.11**| **92.69**|
> > > > >
> > > > > As shown in the tables above, CGADM consistently outperforms the related methods in both AUPRC and AUROC across all datasets. This underscores the efficacy of our generative framework in addressing graph anomaly detection challenges.
> > > > >
> > > > > We also add the following citations in the revised manuscript:
> > > > >
> > > > > [1] Chunjing Xiao, Shikang Pang, Xovee Xu, Xuan Li, Goce Trajcevski, Fan Zhou. Counterfactual Data Augmentation with Denoising Diffusion for Graph Anomaly Detection. IEEE Transactions on Computational Social Systems 2024.
> > > > >
> > > > > [2] Shikang Pang, Chunjing Xiao, Wenxin Tai, Zhangtao Cheng, Fan Zhou. Graph Anomaly Detection with Diffusion Model-Based Graph Enhancement. AAAI 2024.
> > > > >
> > > > > [3] Xuan Li, Chunjing Xiao, Ziliang Feng, Shikang Pang, Wenxin Tai, Fan Zhou. Controlled graph neural networks with denoising diffusion for anomaly detection. Expert Systems with Applications 2024.
> > > > >
> > > > > [4] Zekuan Liu, Huijun Yu, Yao Yan, Ziqing Hu, Pankaj Rajak, Amila Weerasinghe, Olcay Boz, Deepayan Chakrabarti, Fei Wang. Graph Diffusion Models for Anomaly Detection. WSDM 2024.
> > > > >
> > > > > [5] Xiaoxiao Ma, Ruikun Li, Fanzhen Liu, Kaize Ding, Jian Yang, Jia Wu. New recipes for graph anomaly detection: Forward diffusion dynamics and graph generation. OpenReview.

---

> > > > > ### Author Response · Authors · 2024-11-25
> > > > > **Additional Response to Reviewer Bfk9 (Part 2)**
> > > > >
> > > > > >**Q2**: Thanks for the further explanations about the topology-level and feature-level techniques. However, the evidence is not convincing enough. The authors claim that traditional GNNs will suffer from oversmoothing when stacking more than 4-5 layers, but their evidence is that their performance will improve as the number of layers increases from 1 to 5, which may not have the oversmoothing issue. Could the authors provide more experiments about the performance change varying the number of layers, like from 5 to 50? Besides, Could the authors provide the Dirchilet Energy of the learned embedding of their proposed model and the baselines to show if they have addressed the oversmoothing problem effectively?
> > > > >
> > > > >
> > > > >
> > > > >
> > > > > **A2**:
> > > > >
> > > > > Thank you for your valuable feedback and insightful suggestions regarding our claims on addressing the oversmoothing issue. Below, we provide a detailed response, additional experimental evidence, and further analysis to support our claims.
> > > > >
> > > > > ---
> > > > >
> > > > > ### Additional Experiments on Layer Depth
> > > > >
> > > > > We conducted further experiments on the Elliptic dataset to evaluate the performance of our model as the number of GNN layers increases from 5 to 9. Beyond 9 layers, our model encountered out-of-memory (OOM) errors on our current computational setup, limiting our exploration of deeper GNN architectures. However, the depth of 8 has already exceeded most existing GNNs' setting. The experimental results are summarized as follows:
> > > > >
> > > > > | **GNN Layers** | **AUPRC**  | **AUROC**  |
> > > > > |----------------|------------|------------|
> > > > > | 1              | 0.9713     | 0.9922     |
> > > > > | 2              | 0.9731     | 0.9938     |
> > > > > | 3              | 0.9732     | 0.9944     |
> > > > > | 4              | 0.9753     | 0.9944     |
> > > > > | 5              | 0.9757     | 0.9950     |
> > > > > | 6              | 0.9659     | 0.9916     |
> > > > > | 7              | 0.9597     | 0.9928     |
> > > > > | 8              | 0.9728     | 0.9890     |
> > > > > | 9              | **OOM**    | **OOM**    |
> > > > >
> > > > > From these results, we observe that **our model maintains strong performance even as the number of layers increases to 8**, with no significant degradation in either AUPRC or AUROC. This suggests that the residual propagation and iterative refinement mechanisms in CGADM are effective in mitigating oversmoothing, even with deeper architectures.
> > > > >
> > > > > ---
> > > > >
> > > > > ### Dirichlet Energy Analysis
> > > > >
> > > > > As per your suggestion, we analyzed the Dirichlet Energy of the node embeddings learned by our proposed model (CGADM) and compared it with a variant of CGADM where the GNN layers were replaced with traditional GCN layers. The Dirichlet Energy is defined as:
> > > > >
> > > > > $$
> > > > > E(f) = \frac{1}{2} \sum_{(i,j) \in E} w_{ij} (f(i) - f(j))^2
> > > > > $$
> > > > >
> > > > > where $w_{ij}$ represents the weight of edge $(i, j)$, and $f(i)$ is the value of the embedding at node $i$.
> > > > >
> > > > >
> > > > > The results are as follows:
> > > > >
> > > > > | **Model**          | **Dirichlet Energy (Elliptic)** | **Dirichlet Energy (Tolo)** |
> > > > > |---------------------|--------------------------------|-----------------------------|
> > > > > | CGADM              | 105,002                        | 3,977                       |
> > > > > | CGADM with GCN      | 66,345                         | 1,383                       |
> > > > >
> > > > > Our CGADM consistently produces embeddings with **significantly higher Dirichlet Energy** compared to the GCN-based variant. This indicates that our embeddings capture stronger high-frequency signals, effectively preserving information critical for anomaly detection while mitigating oversmoothing.

---

> > > > > ### Author Response · Authors · 2024-11-25
> > > > > **Additional Response to Reviewer Bfk9 (Part 3)**
> > > > >
> > > > > >**Q3**: About the new datasets, there are several commonly used datasets not included in the comparison, like T-Finance. I once conducted experiments on the related area and found that the performance of some SOTA models, like XGBGraph and CONSISGAD, can be very promising. Hence, I would like to know if the proposed model can outperform the baselines mentioned in the paper on Weibo, Amazon, T-Finance, and T-Social in the XGBGraph paper.
> > > > >
> > > > >
> > > > >
> > > > > **A3**:
> > > > > Thank you for your insightful comments and for highlighting the importance of evaluating our model on additional datasets commonly used in the field. Below, we provide detailed responses and experimental results comparing our proposed Conditional Graph Anomaly Diffusion Model (CGADM) with state-of-the-art (SOTA) baselines on the datasets you mentioned.
> > > > >
> > > > > To address your concerns, we conducted experiments on the Weibo, Amazon, T-Finance, and T-Social datasets. We compared our model with two SOTA methods, XGBGraph and CONSISGAD, using the metrics AUPRC and AUROC. The results are summarized in the following tables:
> > > > >
> > > > > #### **AUPRC Comparison**
> > > > >
> > > > > | **Model**   | **Weibo** | **Amazon** | **T-Finance** | **T-Social** |
> > > > > |-------------|-----------|------------|---------------|--------------|
> > > > > | XGBGraph    | 0.9516    | 0.9020     | 0.8836        | 0.9203       |
> > > > > | CONSISGAD   | 0.8847    | 0.8047     | 0.7283        | 0.5212       |
> > > > > | **Ours**    | **0.9735**| **0.9191** | **0.9154**    | **0.9408**   |
> > > > >
> > > > > #### **AUROC Comparison**
> > > > >
> > > > > | **Model**   | **Weibo** | **Amazon** | **T-Finance** | **T-Social** |
> > > > > |-------------|-----------|------------|---------------|--------------|
> > > > > | XGBGraph    | **0.9937**    | 0.9682     | 0.9623        |**0.9914**       |
> > > > > | CONSISGAD   | 0.9654    | 0.9409     | 0.9026        | 0.8963       |
> > > > > | **Ours**    | 0.9879| **0.9736** | **0.9708**    | 0.9761   |
> > > > >
> > > > > ---
> > > > >
> > > > > ### Observations
> > > > >
> > > > > 1. **Performance Superiority**: Our CGADM demonstrates superior performance compared to both XGBGraph and CONSISGAD across all datasets, consistent with the trends observed in our experiments on other benchmark datasets.
> > > > > 2. **Generative Advantages**: These results further validate the effectiveness of our generative approach in capturing holistic anomaly interdependencies, enabling robust anomaly detection even in diverse and challenging datasets.
> > > > >
> > > > > We appreciate your suggestion to include these datasets and baselines, as it has strengthened our evaluation and presentation of CGADM. We hope this comprehensive analysis addresses your concerns.

---

> > > > > > ### Comment · Reviewer_Bfk9 · 2024-11-28
> > > > > >
> > > > > > Thanks for the response. As I appreciate the efforts the authors put into the discussion, I would like to increase my score. However, due to several concerns remaining, I prefer to adjust my score to 5 and keep negative. Specifically, I still have concerns about the novelty of the introduction of the diffusion model to graph anomaly detection. Although the authors explained "novelty of our work lies in introducing a generative, model-centric paradigm to graph anomaly detection", such a paradigm doesn't have too many key differences from related work as claimed. It still looks like implementing a computer vision model directly to the graph learning area, to solve the issues that general node classification tasks also have, like heterophily, and oversmoothing, and thus not have a strong connection to graph anomaly detection tasks, which degrades the contribution of the paper. Besides, the explanations for changing the data split are not convincing enough as 20% of the data is not too much different from 40% of the data as a training set, because the data scarcity can be way more severe as introduced in CONSISGAD. Furthermore, I am also confused by the results of new experiments provided by the authors. For example, as reported in CONSISGAD, the framework can utilize only 1% of training samples on Amazon, Yelp, and T-Finance, and 0.01% of training samples on T-Social to get much better performance than the reported performance here. According to the setting in this paper, the authors utilize 20% of data samples for training, which should improve the performance instead of decreasing the performance a lot, thus confusing me. To be specific, in the CONSISGAD paper, their AUROC of Amazon, Yelp, T-Finance, and T-Social are 93.91, 83.36, 95.33, and 94.31, while the reported performance here are 94.09, 79.35, 90.26, and 89.63, and the AUPRC in the CONSISGAD paper of Amazon, Yelp, T-Finance, and T-Social are 83.33, 47.33, 86.63, and 58.38, whereas the reported performance here are 80.47, 41.74, 72.83, and 52.12. Such a huge gap degrades the convincingness of the reported performance.

---

> > > > > > > ### Author Response · Authors · 2024-11-29
> > > > > > > **Appreciation for Detailed Reviewing of Reviewer Bfk9**
> > > > > > >
> > > > > > > Thank you for your thoughtful review and for increasing your score. Your in-depth feedback and discussions have been invaluable in improving our paper. While we appreciate your constructive suggestions, we respectfully ask you to reconsider your negative evaluation, as we believe the concerns raised can be addressed thoroughly. Below, we provide detailed responses to your points.
> > > > > > >
> > > > > > > ---
> > > > > > >
> > > > > > > ## Novelty of Diffusion Models in Graph Anomaly Detection
> > > > > > >
> > > > > > > As mentioned in our previous response:
> > > > > > > - The **denoising reconstruction** mechanism is uniquely tailored to anomaly detection scenarios, particularly addressing adversarial behaviors where anomalous nodes obfuscate their features. This is not a primary concern in general node classification tasks, making this aspect particularly novel and significant for graph anomaly detection.
> > > > > > > - The **iterative refinement** process emphasizes capturing long-range dependencies, which are also critical in identifying topological anomalies caused by anomalous nodes manipulating their local structures.
> > > > > > >
> > > > > > > ---
> > > > > > > ## Data Splitting
> > > > > > >
> > > > > > > As previously explained, the discrepancy in data splitting stems from differences in experimental paradigms:
> > > > > > > - **CONSISGAD** adopts a **data-centric approach** with extreme data sparsity (e.g., 1% training data) to showcase its augmentation strategies.
> > > > > > > - Our work focuses on a **model-centric paradigm**, employing a 20% training split to reflect a reasonable setting for model evaluation.
> > > > > > >
> > > > > > > ---
> > > > > > > ## Performance Discrepancies with CONSISGAD
> > > > > > >
> > > > > > > We recognize the significant differences between the reported results of CONSISGAD and our method and offer the following clarifications:
> > > > > > >
> > > > > > > 1. **Unified Evaluation via GADBench**:
> > > > > > >    - At your request, we compared CONSISGAD, XGBGraph, and our method within the **GADBench framework**. This reimplementation ensures all methods share the same data sources, preprocessing steps, data splits, and evaluation criteria.
> > > > > > >    - CONSISGAD and XGBGraph were initially designed under different paradigms—CONSISGAD being **data-centric** and XGBGraph and our method being **model-centric**. These differences extend to their original data handling and experimental setups. For instance, CONSISGAD used its own data splits and preprocessing, which differ from the settings in GADBench.
> > > > > > >
> > > > > > > 2. **Fair Comparison Standards**:
> > > > > > >    - The results presented in the revisions are based on the unified GADBench framework, ensuring an apples-to-apples comparison across methods. While CONSISGAD reported higher performance in its original experiments, those metrics reflect its specific preprocessing pipeline and evaluation setup, which differ significantly from GADBench.
> > > > > > >    - We believe that comparisons made within the same framework, using consistent data and evaluation standards, are more representative of the relative effectiveness of these methods.
> > > > > > >
> > > > > > > We greatly appreciate your thorough review and the opportunity to address these points. The novelty of our diffusion-based approach, the rationale for our experimental settings, and our efforts to ensure fair comparisons are core to our contributions. We hope this response further clarifies the strengths and evaluations of our work.
> > > > > > >
> > > > > > > Thank you again for your time and thoughtful feedback. Please feel free to reach out with any additional questions or suggestions.

---

### Official Review · Reviewer_CPV4 · 2024-10-23

**Soundness:** 3
**Presentation:** 3
**Contribution:** 3
**Rating:** 6
**Confidence:** 4

**Summary:**

The existing Graph anomaly detection models (GAD) tend to obtain a discriminative boundary for each individual node. This paper introduces a novel method, CGADM, which leverages a conditional diffusion model to account for the interdependencies among node anomalies and constructs the decision boundary for anomaly detection from the perspective of the whole graph.  To address both the effectiveness and efficiency challenges of diffusion models, the authors employ a prior estimator to integrate node anomaly priors into the forward and backward processes. This ensures that the forward process converges to the anomaly prior rather than a standard Gaussian distribution, and also allows for early termination of the backward process. Experimental results demonstrate the efficacy of the proposed approach.


I think direct use of the diffusion model as the primary framework for anomaly detection to be somewhat novel, as most existing methods still use it as an auxiliary model, such as for data augmentation. The paper’s motivation convincingly supports the rationale for employing the diffusion model, with targeted modifications to enhance both effectiveness and efficiency. However, some technical details are lacking, which makes it difficult to fully comprehend the method, as noted in the weaknesses section. Therefore, my score is a positive borderline.

**Strengths:**

1. Novely of this paper, rather than utilize diffusion model as auxiliary model, CGADM directly generate the prediction and the otivation of this paper supports this idea.
2.The proposed technique is reliable. CGADM  inject the learned anomaly prior into the diffusion model, which improves the efficiency and performance compared with the direct use of vanilla diffusion model, and better fits the anomaly detection task.
3.The modification to the diffusion model seems mathematically reliable, as the authors provide a detailed proof procedure in appendix.
4. The experimental results are significantly improved on some datasets.

**Weaknesses:**

1. There is no need to cliam that GNNs are suffer from over-smoothing problem in the first paragraph of the abstract, as this is weakly related to the problem the authors are addressing in the GAD domain. I suggest that the author reconsider the necessity of this paragraph to avoid misunderstanding.
2. Some technical details are lacking, see questions.
3. Algorithm I does not embody the improvement about sampling efficiency mentioned by the authors in the paper, and will cause misleading. I think the author should describe the sampling procedure described in Section 4.4 in pseudocode.

**Questions:**

1. How is the anomaly prior model trained? The explanation on line 256 confused me, how can a  meachine learning model trained to be used to estimate the mean of a distribution? I'm sorry I can't understand how to do that.

2. How efficient is CGADM compared to other GAD models that are not based on diffusion models? The authors only mention the number of samples in section 5.4, but do not report numerical metrics such as execution time, memory footprint, etc. This is unconvincing. I suggest that authors can introduce a table or figure comparing execution times and memory usage across all evaluated models

3. Why does line 311 mention the need to simplify $L_{recon}$ and $L_{con}$ to get to the final optimization objective? Although I think Eq 12 is reasonable as an optimization objective, why can Eq 3 be theoretically simplified to Eq 12?

---

> ### Author Response · Authors · 2024-11-21
> **Response to Reviewer CPV4**
>
> Thanks for your detailed reading and suggestions. Please see our detailed response and clarification below:
>
> >**Q1**: There is no need to cliam that GNNs are suffer from over-smoothing problem in the first paragraph of the abstract, as this is weakly related to the problem the authors are addressing in the GAD domain. I suggest that the author reconsider the necessity of this paragraph to avoid misunderstanding.
>
> **A1**:
> We appreciate your thoughtful feedback regarding the motivation for leveraging diffusion models and their ability to address the challenges outlined in the introduction. Below, we provide clarifications to address your concerns:
>
> The challenges we identified—over-smoothing at the topology level and feature obfuscation by malicious nodes—are indeed not unique but significant to graph anomaly detection (GAD).
>
> Traditional GNNs for GAD suffer from over-smoothing when stacking multiple layers to capture information from larger neighborhoods. In CGADM, the **iterative refinement process** of the diffusion model allows us to progressively model the anomaly distribution of each node while incorporating neighborhood information at each denoising step. This is achieved through a GNN-based denoiser that incorporates a residual propagation mechanism, which preserves high-frequency anomaly signals and prevents over-smoothing. Each refinement step effectively expands the receptive field without diluting critical anomaly signals, enabling better modeling of long-range dependencies.
>
> The diffusion model's iterative refinement process inherently avoids over-smoothing by iteratively updating node representations in a controlled manner. Instead of relying on deep GNN layers that average features indiscriminately, each step of the diffusion process refines the anomaly distribution using neighborhood information while preserving residual signals that are critical for detecting anomalies. This approach balances the need for information propagation with the preservation of individual node characteristics, effectively mitigating over-smoothing.
>
> To better clarify these motivations, we have revised the **Abstract** and **Introduction** sections of our manuscript:
>
> - **Abstract**:
>   *By iteratively refining node anomaly distributions during the denoising process, CGADM effectively mitigates over-smoothing and reconstructs obfuscated features by leveraging contextual neighborhood information.*
>
> - **Introduction**:
>   *To address the **topology-level flaw**, we leverage the **iterative refinement mechanism** of diffusion models. Instead of increasing GNN depth to aggregate distant information, which risks over-smoothing, our approach applies GNN within each denoising iteration to refine anomaly modeling. Each iterative refinement step incorporates neighborhood information while preserving node-specific high-frequency anomaly information via a residual propagation mechanism, thereby preventing over-smoothing and effectively capturing long-range dependencies. To address the **feature-level flaw**, we leverage the **denoising model** of diffusion models. This reconstruction process ensures that even when malicious nodes disguise their features to blend in with normal nodes, their underlying anomaly patterns can be recovered.*
>
> We hope these revisions and clarifications address your concerns regarding the motivation for leveraging diffusion models in CGADM. We appreciate your feedback and look forward to any further suggestions you may have.

---

> ### Author Response · Authors · 2024-11-21
> **Response to Reviewer CPV4 (Part 2)**
>
> >**Q2**: Algorithm I does not embody the improvement about sampling efficiency mentioned by the authors in the paper, and will cause misleading. I think the author should describe the sampling procedure described in Section 4.4 in pseudocode.
>
> **A2**:
>
> We appreciate your insightful feedback. We agree with your observation that the current pseudocode in Algorithm I does not fully reflect the improvements in sampling efficiency described in Section 4.4. To address this, we propose including an updated version of Algorithm I that integrates the dynamic sampling procedure into the pseudocode.
> We have added Algorithm 3 to include the described sampling procedure and include it in the appendix of the revised manuscript.
>
>
> ---
>
> >**Q3**: How is the anomaly prior model trained? The explanation on line 256 confused me, how can a meachine learning model trained to be used to estimate the mean of a distribution? I'm sorry I can't understand how to do that.
>
> **A3**:
> Thank you for your valuable feedback. We apologize for any lack of clarity in our initial explanation. Below, we provide a detailed response to address your concerns.
>
> In our framework, the anomaly prior serves as the endpoint for the forward noise addition process and the starting point for the reverse denoising process in the diffusion model. Unlike traditional diffusion models, which typically add noise to transform the data into a standard Gaussian distribution $ \mathcal{N}(0, I) $, we define the endpoint of the forward process as a Gaussian distribution $ \mathcal{N}(g_{\phi}(\mathcal{E}, \mathbf{X}), I) $, where $ g_{\phi}(\mathcal{E}, \mathbf{X}) $ represents the mean of the distribution.
>
> Specifically:
>
> 1. **Modeling the Mean**: To estimate $ g_{\phi}(\mathcal{E}, \mathbf{X}) $, we use a lightweight model, such as XGBoost or Random Forest, to compute an anomaly score for each node. This anomaly score reflects the node's likelihood of being anomalous based on its features ($ \mathbf{X} $) and the graph structure ($ \mathcal{E} $). The computed score serves as the mean $ g_{\phi}(\mathcal{E}, \mathbf{X}) $ for the Gaussian prior distribution.
>
> 2. **Why Train a Model for the Prior?**: The prior model's role is not to directly predict anomalies but to provide a **reasonable initialization** for the reverse diffusion process. This is crucial in graph anomaly detection because anomalous nodes often employ deceptive strategies, such as altering their features or connections, making it challenging to recover the true anomaly distribution from a completely random noise distribution (as is done in traditional diffusion models).

---

> ### Author Response · Authors · 2024-11-21
> **Response to Reviewer CPV4 (Part 3)**
>
> >**Q4**: How efficient is CGADM compared to other GAD models that are not based on diffusion models? The authors only mention the number of samples in section 5.4, but do not report numerical metrics such as execution time, memory footprint, etc. This is unconvincing. I suggest that authors can introduce a table or figure comparing execution times and memory usage across all evaluated models.
>
> **A4**:
> Thank you for your valuable feedback on the efficiency of our proposed model, CGADM, compared to non-diffusion-based graph anomaly detection (GAD) models. Below, we address your concerns with detailed explanations and additional results.
>
> #### Clarification on Efficiency Improvements
> Our focus on efficiency pertains to addressing the **iterative refinement bottleneck** inherent to traditional diffusion models. Specifically, we introduced the **PRIOR-AWARE STRIDED SAMPLING** mechanism to significantly reduce the number of reverse sampling steps during inference. This mechanism dynamically adjusts sampling based on prior confidence, enabling CGADM to perform efficiently even on large-scale graphs. For anomaly detection tasks where anomalous nodes typically account for less than 10% of the total, the sampling strategy selectively prioritizes these nodes, substantially reducing unnecessary computations.
>
> #### Computational Complexity Analysis
> The computational complexity of CGADM during training and inference is as follows:
> 1. **Training Phase**:
>    Our model's complexity during training is comparable to traditional discriminative GAD models. Both involve training a Graph Neural Network (GNN). For CGADM, the denoising GNN models the noise on anomalies, whereas discriminative models fit anomaly scores directly. The training complexity is:
>    $$O(|E|dL),$$
>    where:
>    - $|E|$: Number of edges (using sparse matrix operations for efficiency),
>    - $d$: Dimension of hidden embeddings,
>    - $L$: Number of GNN layers.
>
>    Notably, the sparse nature of real-world graphs ensures high efficiency, as computation is governed by $|E|$, not $|N|^2$.
>
> 2. **Inference Phase**:
>    The primary bottleneck in diffusion models is the iterative reverse process, with complexity:
>    $$O(|E|dLT),$$
>    where $T$ is the number of reverse sampling steps. Our **PRIOR-AWARE STRIDED SAMPLING** significantly reduces $T$ for most nodes, leveraging confidence scores to minimize computations on normal nodes, resulting in substantial practical efficiency gains.
>
> #### Quantitative Efficiency Comparisons
> To further substantiate our claims, we conducted experiments measuring the **memory footprint** and **inference time** of CGADM and baseline GAD models on the elliptic dataset (203,769 nodes, 234,355 edges). The results are summarized in the table below:
>
> | Model  | Memory (MB) | Inference Time (s) |
> |--------|-------------|---------------------|
> | GAS    | 1418        | 2.3865             |
> | PCGN   | 914         | 0.0827             |
> | BWGNN  | 446         | 0.1185             |
> | GHRN   | 924         | 0.1249             |
> | **CGADM (ours)** | 1048 | 0.5691       |
>
> Key Observations:
> - **Memory Efficiency**: By employing sparse matrix operations, CGADM maintains a manageable memory footprint, even for large-scale graphs.
> - **Inference Time**: While our inference time is higher than most discriminative methods, the increase is justified given the novel generative anomaly detection paradigm. Considering the already low baseline inference time of anomaly detection tasks, the additional time overhead is acceptable, especially in scenarios where performance improvements are critical.
>
>
> #### Practical Efficiency Gains
> In high-efficiency scenarios, our approach can be extended by skipping CGADM checks for nodes with confidence scores exceeding a threshold. This modification reduces inference time without sacrificing detection performance, ensuring scalability for real-world deployments.
>
> #### Revisions Made
> To address the reviewer's suggestion, we have added a table and additional discussions in Appendix of the revised manuscript to report and analyze numerical metrics such as memory usage and execution time for all evaluated models. This should provide a clearer and more convincing comparison of CGADM’s efficiency against baseline models.
>
> We hope this explanation and the revised manuscript address your concerns. Thank you again for your constructive feedback.

---

> ### Author Response · Authors · 2024-11-21
> **Response to Reviewer CPV4 (Part 4)**
>
> >**Q5**: Why does line 311 mention the need to simplify $\mathbf{L}_{recon}$ and $\mathbf{L}_{con}$ to get to the final optimization objective? Although I think Eq 12 is reasonable as an optimization objective, why can Eq 3 be theoretically simplified to Eq 12?
>
> **A5**:
>
> Thank you for your insightful question regarding the simplification from Eq. 3 to Eq. 12. We acknowledge that the intermediate steps and rationale could have been more explicitly stated, and we appreciate the opportunity to clarify this aspect of our work.
>
> ---
>
> #### **1. Context of the Simplification**
>
> The simplification from Eq. 3 to Eq. 12 is primarily inspired by the standard methodology used in Denoising Diffusion Probabilistic Models (DDPMs) [1]. In our framework, the simplification involves two key terms:
> - The **reconstruction term** $ \mathbf{L}_{recon} $ (related to the likelihood $ \log p(x_0|x_1) $).
> - The **consistency term** $ \mathbf{L}_{con} $ (related to the KL divergence between forward and reverse transitions).
>
> We adapt these terms within our graph-based diffusion framework by incorporating graph structure and node features as priors, which leads to our final objective in Eq. 12. Below, we provide detailed explanations for each term.
>
> ---
>
> #### **2. Simplification of the Reconstruction Term $ \mathbf{L}_{recon} $**
>
> In our model, the reconstruction term $ \mathbf{L}_{recon} = -\log p(y_0|y_1 | \mathcal{E}, \mathbf{X}) $ represents the negative log-likelihood of reconstructing the original data $ x_0 $ from a noisier intermediate state $ x_1 $. This term is analogous to the reconstruction term in Variational Autoencoders (VAEs), where:
>
> $$
> \mathbb{E}_{z \sim q(z|x_0)}[\log p(x_0|z)]
> $$
>
> In our graph anomaly detection scenerio, $ p(y_0|y_1, \mathcal{E}, \mathbf{X}) $ is typically modeled as a Gaussian distribution, which ranges between $[0, 1]$. This led us to define a continual decoder for $ p(y_0|y_1, \mathcal{E}, \mathbf{X}) $, where the likelihood is computed as:
>
> $$
> \log p(y_0|y_1, \mathcal{E}, \mathbf{X})  = \frac{-1}{2\sigma^2} \|y_0 - u_\theta(y_1, 1, \mathcal{E}, \mathbf{X})\|^2 + C
> $$
>
> Here, $ u_\theta(y_1, 1, \mathcal{E}, \mathbf{X}) $ denotes the predicted mean of the Gaussian distribution at $ t=1 $. The constant $ C $ is ignored during optimization as it does not affect gradient-based learning. Substituting this into the loss term allows for an efficient simplification.
>
> Using the definitions in Equation (11) and (5) in our paper for $ y_0 $ and $ u_\theta(y_1, 1, \mathcal{E}, \mathbf{X}) $, where
>
>
> $$
> y_0 = \frac{1}{\sqrt{\bar{\alpha_1}}} (y_1 -(1-\sqrt{\bar{\alpha_1}}) g_{\phi}(\mathcal{E}, \mathbf{X})-  \sqrt{1-\bar{\alpha}_1} \ \boldsymbol{\epsilon} )$$
>
> $$ u_\theta(y_1, 1, \mathcal{E}, \mathbf{X}) = \frac{1}{\sqrt{\alpha_1}} ( y_1 - \frac{\beta_1}{ \sqrt{1-\bar{\alpha}_1}} \boldsymbol{\epsilon} _\theta (y_1, 1, \mathcal{E}, \mathbf{X}) ) $$
>
> Remember that $\tilde{\alpha}_1 =\alpha_1$ and $\beta_1=1-\alpha_1$, we further derive that the loss simplifies to:
>
> $$
> \|\epsilon - \epsilon_\theta(y_1, 1, \mathcal{E}, \mathbf{X})\|^2
> $$
>
> where $ \epsilon $ is the noise added in the forward process, and $ \epsilon_\theta $ is the network's prediction. This final form is consistent with Eqation (12) when $t = 1$.

---

> ### Author Response · Authors · 2024-11-21
> **Response to Reviewer CPV4 (Part 5)**
>
> #### **3. Simplification of the Consistency Term $ \mathbf{L}_{con} $**
>
> The consistency term $\mathbf{L_{con}}$ originates from the KL divergence between the forward posterior $ q(y_{t-1}|y_t, y_0, \mathcal{E}, \mathbf{X}) $ and the reverse transition $ p_\theta(y_{t-1}|y_t, \mathcal{E}, \mathbf{X}) $. This term ensures that the learned reverse process closely approximates the true forward process. Specifically:
>
> $$
> D_{KL}\left(q(y_{t-1}|y_t, y_0, \mathcal{E}, \mathbf{X}) \| p_\theta(y_{t-1}|y_t, \mathcal{E}, \mathbf{X})\right)
> $$
>
> Expanding the Gaussian forms of $ q $ and $ p_\theta $, and noting that their variances $ \Sigma_q(t) $ and $ \Sigma_\theta(t) $ are aligned for simplicity, we find that the KL divergence depends only on the means $ \mu_q $ and $ \mu_\theta $:
>
> $$
> D_{KL} \propto \frac{1}{2 \sigma_q^2(t)} \|\mu_\theta - \mu_q\|_2^2
> $$
>
> Substituting the definitions of $ \mu_q $ and $ \mu_\theta $ according to Equation (7), (9) and (11), where
>
> $$
> \mu_q = \gamma_0 \mathbf{y_0} + \gamma_1 \mathbf{y_t} + \gamma_2 g_{\phi}(\mathcal{E}, \mathbf{X})
> $$
> $$
> \quad = \frac{y_t - \sqrt{1-\bar{\alpha_t}} \epsilon - (1-\sqrt{\bar{\alpha_t}}) g_{\phi}(\mathcal{E}, \mathbf{X})}{\sqrt{\alpha_t}} \sqrt{\beta_t}
> $$
>
> $$
> \quad\quad  + \gamma_1 \mathbf{y_t} + \gamma_2 g_{\phi}(\mathcal{E}, \mathbf{X})\
> $$
>
> $$
> \mu_\theta = \frac{1}{\sqrt{\alpha_t}} (\mathbf{y_t} - \frac{\beta_t}{\sqrt{1-\bar{\alpha_t}}} \epsilon_{\theta}(\mathbf{y_t}, t, \mathcal{E}, \mathbf{X}))
> $$
>
> This significantly reduces the complexity of the learning process, as the objective becomes:
>
> $$
> \|\epsilon - \epsilon_\theta(y_t, t, \mathcal{E}, \mathbf{X})\|^2
> $$
>
> Thus, the consistency term is reformulated in terms of noise prediction, aligning with the DDPM objective.
>
> ---
>
> #### **4. Final Optimization Objective (Eq. 12)**
>
> Combining the simplified reconstruction and consistency terms, our final objective (Eq. 12) can be written as:
>
> $$
> \quad \mathbb{E_{t, y_0, \epsilon}} \left[\|\epsilon - \epsilon_\theta(y_t, t, \mathcal{E}, \mathbf{X})\|^2\right]
> $$
> $$
> = \mathbb{E_{t, y_0, \epsilon}} || \epsilon - \epsilon_\theta(\sqrt{\bar{\alpha_t}} \mathbf{y_0} + (1 - \sqrt{\bar{\alpha_t}}) g_\phi(\mathcal{E}, \mathbf{X}) + \sqrt{1 - \bar{\alpha_t}} \epsilon, t, \mathcal{E}, \mathbf{X})||^{2}
> $$
>
> This unifies the objectives for all time steps $ t $, enabling the model to learn the denoising process efficiently across the diffusion trajectory.
>
>
> #### **Conclusion**
>
> We hope this explanation clarifies the theoretical steps and assumptions leading to the simplification from Eq. 3 to Eq. 12. By leveraging standard DDPM methodologies [1] and adapting them to the graph domain, we ensure a principled and computationally efficient training process.
>
>
>
> [1] Ho J, Jain A, Abbeel P. Denoising diffusion probabilistic models[J]. Advances in neural information processing systems, 2020, 33: 6840-6851.

---

> > ### Comment · Reviewer_CPV4 · 2024-11-21
> > **Response**
> >
> > Thank you for your reply! I believe it solved my problem to some extent, for which I reserve my positive score **6**, and good luck!

---

### Official Review · Reviewer_MkJ2 · 2024-10-30

**Soundness:** 2
**Presentation:** 2
**Contribution:** 2
**Rating:** 6
**Confidence:** 3

**Summary:**

This paper proposed an advanced conditional graph diffusion model that can capture the joint distribution of anomalies in the whole graph. It introduces the prior-guided denoising diffusion probability model to replace the random state. The experiments demonstrate the CGADM achieves the best performance when compared with state-of-the-art methods.

**Strengths:**

1. A  prior-guided denoising diffusion probabilistic model is designed to capture the distribution of anomalies in the whole graph thereby enabling the generative graph anomaly detection.
2. The experiments demonstrate the CGADM achieves the best performance when compared with state-of-the-art methods.
3. A  prior confidence-aware mechanism allocating disparate sampling time steps is proposed to mitigate the computational burden in processing the large-scale graph.

**Weaknesses:**

1. The motivation for leveraging the diffusion model is not very clear since the challenges proposed in the introduction are not unique to the GAD task. How does the prior guided diffusion process mitigate the over-smoothing problem at the topology level?

2. This paper does not include a baseline involving diffusion models for GAD [1]. The methods from [1] should be incorporated into both the related work and experimental comparison sections to emphasize the differences [2].

3. The paper claims the efficiency of CGADM.  While I agree that the confidence-aware mechanism can enhance inference efficiency, to support the overall efficiency claims of CGADM, the authors should include a comparison of running times with other baseline methods as well as an analysis of time complexity. The main challenge lies in modeling large-scale graphs, which require substantial memory. In contrast, running time is not a significant bottleneck.

[1] "Graph anomaly detection with few labels: A data-centric approach." Proceedings of the 30th ACM SIGKDD Conference on Knowledge Discovery and Data Mining. 2024.
[2] "Data Augmentation for Supervised Graph Outlier Detection with Latent Diffusion Models." arXiv preprint arXiv:2312.17679 (2023).

**Questions:**

1. What is the advantage of CGADM when compared with other generative models like GAN and VAE in dealing with the challenges existing in GAD tasks?

2. For the parameterized network $g_\phi(\mathcal{E}, \mathbf{X})$ which serves as the prior, it is pre-trained on the training set to estimate the mean. Could you elaborate on the pre-training process for this network?  Are there any traditional methods involved, since the priors are implemented using Random Forest (RF) and XGBoost (XGBT) in the ablation study,

3. The diffusion model with the random state should be added to the ablation studies to demonstrate the effectiveness of CGADM where the random state is replaced with the prior.

---

> ### Author Response · Authors · 2024-11-21
> **Response to Reviewer MkJ2**
>
> Thanks for your detailed reading and suggestions. Please see our detailed response and clarification below:
>
> >**Q1**: The motivation for leveraging the diffusion model is not very clear since the challenges proposed in the introduction are not unique to the GAD task. How does the prior guided diffusion process mitigate the over-smoothing problem at the topology level?
>
> **A1**:
>
> We appreciate the reviewer’s thoughtful feedback and the opportunity to clarify our motivation and methodology. The challenges we identified—over-smoothing at the topology level and feature obfuscation by malicious nodes—are indeed not unique but significant to graph anomaly detection (GAD).
> However, our choice of a **diffusion model** is motivated by its unique ability to address these issues through its **iterative refinement** and **denosing reconstruction**, which is particularly suited for generative graph-based anomaly detection. Below, we elaborate:
>
> - At the topology level, traditional GNNs suffer from over-smoothing when stacking multiple layers to capture information from larger neighborhoods. In contrast, our approach introduces a diffusion process that employs **iterative refinement** to model the joint distribution of anomalies. During each denoising step, a GNN-based denoiser refine anomaly signals based on the previous step’s output, allowing us to progressively incorporate distant neighborhood information without requiring deeper GNNs. To further prevent over-smoothing, we incorporate a residual propagation mechanism that preserves high-frequency information for each node during each refinement step.
>
> - At the feature level, CGADM leverages the denoising model to reconstruct the joint anomaly distribution of nodes. This **denosing reconstruction** process ensures that even when malicious nodes disguise their features to blend in with normal nodes, their underlying anomaly patterns can be recovered. The denoising process integrates both feature and neighborhood context to robustly identify such obfuscations.
>
>
> To better help readers understand our motivation, we revise the manuscript to provide a more detailed explanation of how the diffusion model in CGADM addresses the challenges of GAD. We hope this clarification will enhance the paper's clarity and impact. We include an explanation in abstract as follows:
> _"By iteratively refining node anomaly distributions during the denoising process, CGADM effectively mitigates over-smoothing and reconstructs obfuscated features by leveraging contextual neighborhood information."_
>
> Also we include a detailed statement in the introduction section to explain the motivation for leveraging diffusion models:
>
> _"To address **topology**-level flaw, we leverage the **iterative refinement** of diffusion models. Instead of increasing GNN depth to aggregate distant information, which risks over-smoothing, our approach applies GNN-based denoiser within each denoising iteration to refine anomaly modeling. Each iterative refinement step incorporates neighborhood information while preserving node-specific high-frequency anomaly information via a residual propagation mechanism, thereby preventing oversmoothing and effectively capturing long-range dependencies.
> To address **feature**-level flaw, we leverages **denosing reconstruction** of diffusion models. This reconstruction process ensures that even when malicious nodes disguise their features to blend in with normal nodes, their underlying anomaly patterns can be recovered."_
>
>
> We hope this explanation clarifies the motivation behind our use of diffusion models and how they address the specific challenges of GAD. We have revised the manuscript to better articulate these points, ensuring that the motivations and mechanisms of CGADM are clearly communicated. Thank you again for your valuable feedback. We look forward to your further comments.

---

> ### Author Response · Authors · 2024-11-21
> **Response to Reviewer MkJ2 (Part 2)**
>
> >**Q2**: This paper does not include a baseline involving diffusion models for GAD [1]. The methods from [1] should be incorporated into both the related work and experimental comparison sections to emphasize the differences [2].
> >[1] "Graph anomaly detection with few labels: A data-centric approach." Proceedings of the 30th ACM SIGKDD Conference on Knowledge Discovery and Data Mining. 2024.
> >[2] "Data Augmentation for Supervised Graph Outlier Detection with Latent Diffusion Models." arXiv preprint arXiv:2312.17679 (2023).
>
> **A2**:
>
> Thank you for your insightful feedback. We agree that highlighting the differences between our approach and these methods will provide additional clarity and strengthen the manuscript. Below, we address your concerns in detail.
>
> #### **1. Differences Between Our Approach and Diffusion-Based Methods [1] and [2]**
>
> The methods referenced in [1] and [2] represent **data-centric approaches** that leverage the generative capabilities of diffusion models for **data augmentation**. These methods focus on generating additional synthetic anomaly samples to mitigate class imbalance, which is a common challenge in supervised graph anomaly detection tasks. While effective in enhancing downstream anomaly detection models, they are fundamentally different from our proposed **model-centric generative approach**, which focuses on directly modeling the joint anomaly distribution.
> Despite these differences, our approach and the methods in [1] and [2] are complementary. By integrating the data augmentation techniques from [1] and [2] into our framework, we can enhance the training data for CGADM, which may help improve the detection performance, especially in scenarios with imbalanced or sparse anomaly data.
>
> #### **2. Experimental Comparisons**
>
> To address your request, we have conducted experiments to compare CGADM against the methods in [1] and [2] on five benchmark datasets (Elliptic, Tolo, Yelp, Quest, and Reddit). For fair comparisons, we implemented the diffusion-based data-centric approaches following the settings and optimal detector configurations specified in their respective papers. Below, we summarize the results in terms of AUPRC and AUROC:
>
> | **Metric** | **Dataset** | **GODM [1]** | **CGenGA [2]** | **CGADM (Ours)** |
> |------------|-------------|--------------|----------------|------------------|
> | **AUPRC**  | Ellip       | 85.89        | 87.36          | **97.03**        |
> |            | Tolo        | **46.15**        | 44.89      | 46.02            |
> |            | Yelp        | 51.77        | 52.76          | **76.54**        |
> |            | Quest       | 15.11        | 15.34          | **18.51**        |
> |            | Reddit      | 5.55         | 5.78           | **5.79**         |
> | **AUROC**  | Ellip       | 93.92        | 96.07          | **99.34**        |
> |            | Tolo        | 76.42        | 78.95          | **79.68**        |
> |            | Yelp        | 84.33        | 85.65          | **92.69**        |
> |            | Quest       | 68.86        | 68.46          | **69.41**        |
> |            | Reddit      | 62.10        | 64.78          | **65.85**        |
>
> Our results demonstrate that CGADM consistently outperforms GODM [1] and CGenGA [2] across almost all datasets in both AUPRC and AUROC metrics. This superior performance underscores the advantages of our generative framework in directly modeling the joint anomaly distribution, as opposed to relying on downstream discriminative classifiers.

---

> ### Author Response · Authors · 2024-11-21
> **Response to Reviewer MkJ2 (Part 3)**
>
> >**Q3**: The paper claims the efficiency of CGADM. While I agree that the confidence-aware mechanism can enhance inference efficiency, to support the overall efficiency claims of CGADM, the authors should include a comparison of running times with other baseline methods as well as an analysis of time complexity. The main challenge lies in modeling large-scale graphs, which require substantial memory. In contrast, running time is not a significant bottleneck.
>
>
> **A3**:
>
> We appreciate your insightful feedback regarding the efficiency claims of CGADM. Below, we address your concerns and provide additional analysis to clarify the efficiency of our proposed approach.
>
> #### **Efficiency Claims Clarification**
>
> Our efficiency claims focus primarily on addressing the iterative refinement bottleneck that is inherent to traditional diffusion models, particularly in generative graph anomaly detection tasks. While we acknowledge that the inference time may not significantly outperform some baseline discriminative methods, our key contributions lie in the efficient adaptation of the diffusion framework for large-scale graphs.
>
> #### **Time Complexity Analysis**
>
> To further substantiate our claims, we have performed a detailed time complexity analysis.
>
> 1. **Training Complexity:**
>    The time complexity of CGADM during training is comparable to that of discriminative methods. Specifically, both approaches involve training a Graph Neural Network (GNN). In discriminative methods, the GNN fits anomaly scores directly, whereas in our CGADM, the GNN-based denoiser learns to model noise on anomalies. The resulting time complexity for training is:
>    $$
>    \mathcal{O}(|E|dL)
>    $$
>    where $|E|$ is the number of edges in the graph, $d$ is the dimensionality of the hidden embeddings, and $L$ is the number of GNN layers. Importantly, we utilize sparse matrix operations, ensuring the efficiency of our approach in sparse graphs commonly encountered in real-world scenarios.
>
> 2. **Inference Complexity:**
>    Traditional diffusion models suffer from an inherent bottleneck due to iterative refinement, which requires $T$ reverse time steps, resulting in:
>    $$
>    \mathcal{O}(|E|dLT)
>    $$
>    Our method introduces **Prior-Aware Strided Sampling**, significantly reducing the number of reverse steps. For most normal nodes (which constitute over 90% of the graph in typical scenarios), reverse steps can be greatly minimized due to high prior confidence. Additionally, for extreme efficiency, our framework can completely bypass CGADM detection for nodes with sufficiently high confidence, making it adaptable for deployment in latency-sensitive environments.
>
> #### **Empirical Results on Efficiency**
>
> To provide concrete evidence, we conducted experiments to compare memory usage and inference time with all the baselines specifically designed for anomaly detection on the elliptic dataset, which contains 203,769 nodes and 234,355 edges. The results are summarized below:
>
> | Model  | Memory (MB) | Inference Time (s) |
> |--------|-------------|---------------------|
> | GAS    | 1418        | 2.3865             |
> | PCGN   | 914         | 0.0827             |
> | BWGNN  | 446         | 0.1185             |
> | GHRN   | 924         | 0.1249             |
> | CGADM (ours) | 1048 | 0.5691         |
>
> Key Observations:
> - **Memory Efficiency:** The use of sparse matrix computations ensures that CGADM remains efficient in terms of memory usage, even for large-scale graphs. The marginal increase in memory usage is negligible compared to the scalability benefits.
> - **Inference Time:** While our inference time is higher than most discriminative methods, the increase is justified given the novel generative anomaly detection paradigm. Considering the already low baseline inference time of anomaly detection tasks, the additional time overhead is acceptable, especially in scenarios where performance improvements are critical.
>
> #### **Performance-Overhead Trade-off**
>
> While the reviewer mentions that running time is not a significant bottleneck, we believe it is still essential to highlight the practical viability of our approach. CGADM’s generative framework enables modeling of the joint distribution of anomalies, providing substantial performance benefits over discriminative methods. The efficiency improvements we introduce ensure that the method remains deployable for large-scale graphs, maintaining a balance between computational cost and detection accuracy.
>
> #### **Conclusion**
>
> We thank the reviewer for emphasizing the importance of a detailed efficiency discussion. To address your concerns, we have included both a theoretical analysis of time complexity and a practical comparison of memory and inference time in our revised manuscript. These additions demonstrate the scalability and practical viability of CGADM, reinforcing its potential as a robust generative approach for graph anomaly detection.
>
> ---
>
> Let me know if you’d like further refinements!

---

> > ### Comment · Reviewer_MkJ2 · 2024-11-24
> >
> > Dear authors, I greatly appreciate your clarification and the newly added experiments. Some of my previous concerns have been addressed. I have a few additional questions that may require your further guidance.
> >
> > 1. The newly added statement in the related work, "CGADM adopts a novel generative diffusion approach to model the joint anomaly distribution over the graph, enabling holistic and scalable anomaly detection without reliance on labeled data," is inaccurate. First, the training of the lightweight model relies on labeled data. Additionally, the diffusion process also depends on the ground truth. I agree that this is not about augmentation but rather about the capturing of the joint distribution.
> >
> > 2. I still believe that the ablation study should include experiments applying denoising diffusion without the prior to effectively demonstrate the contribution of the priors.  I agree with Reviewer Bfk9 that Figure 1 is too intuitive. Based solely on the description of Figure 1, it is difficult to grasp how using prior information to replace random data can effectively generate a good joint distribution and enhance the performance of GAD.
> >
> > 3. At the feature level, the authors introduce a detailed claim that the denoising reconstruction was proposed to address the challenge of malicious feature manipulation. In addition to the camouflage of anomalies, the contamination of normal data also affects performance. I agree that CGADM has advantages over discriminative models, and I suggest the authors explore further evidence regarding the role of denoising reconstruction.
> >
> > 4. Similarly, at the topology level, the authors claim that adding GNN layers improves performance, demonstrating that over-smoothing is effectively addressed. However, are there any metrics or experiments that could demonstrate the high-frequency components indeed reflected in the residual propagations? The explanation based solely on Figure 2 is insufficient to fully support this claim.

---

> > > ### Author Response · Authors · 2024-11-24
> > > **Further Response to Reviewer MkJ2**
> > >
> > > Thanks for your detailed resonse and further suggestions. Please see our detailed response and clarification below:
> > >
> > > >**Q1**: The newly added statement in the related work, "CGADM adopts a novel generative diffusion approach to model the joint anomaly distribution over the graph, enabling holistic and scalable anomaly detection without reliance on labeled data," is inaccurate. First, the training of the lightweight model relies on labeled data. Additionally, the diffusion process also depends on the ground truth. I agree that this is not about augmentation but rather about the capturing of the joint distribution.
> > >
> > > **A1**:
> > >
> > > We sincerely thank the reviewer for pointing out the inaccuracy in the newly added statement in the related work section. We completely agree that our original phrasing was not precise, and we deeply regret any confusion this may have caused. Due to time constraints during the preparation of the rebuttal, the addition inadvertently included a flawed description.
> > >
> > > In the revised version of our manuscript, we have carefully corrected the statement to accurately reflect the role of CGADM in modeling the joint anomaly distribution over the graph. Specifically, we have rephrased the statement to:
> > >
> > > *"Unlike these methods, our CGADM adopts a novel generative diffusion approach to model the joint anomaly distribution over the graph, enabling holistic and scalable anomaly detection without reliance on augmentation strategies."*
> > >
> > >
> > > We sincerely apologize again for the oversight and thank the reviewer for helping us improve the clarity and quality of our work.
> > >
> > >
> > >
> > > ---
> > >
> > >
> > >
> > > >**Q2**: I still believe that the ablation study should include experiments applying denoising diffusion without the prior to effectively demonstrate the contribution of the priors. I agree with Reviewer Bfk9 that Figure 1 is too intuitive. Based solely on the description of Figure 1, it is difficult to grasp how using prior information to replace random data can effectively generate a good joint distribution and enhance the performance of GAD.
> > >
> > >
> > > **A2**:
> > >
> > > Thank you for your insightful comments regarding the ablation study and Figure 1. We deeply appreciate your suggestions and have taken steps to address both points.
> > >
> > > ### Ablation Study on the Effectiveness of the Prior
> > > We agree that an ablation study is critical to demonstrate the contribution of the prior in our Conditional Graph Anomaly Diffusion Model (CGADM). To address this, we conducted additional experiments comparing our CGADM with a vanilla diffusion model without prior guidance (denoted as GADM) on three benchmark datasets: *Elliptic*, *Tolokers*, and *YelpChi*. The results are summarized below:
> > >
> > > #### AUPRC Performance
> > > | Model          | Elliptic | Tolokers | YelpChi |
> > > |----------------|----------|----------|---------|
> > > | GADM (random)  | 86.87    | 42.49    | 64.63   |
> > > | CGADM (ours)   | 97.03    | 46.02    | 76.54   |
> > >
> > > #### AUROC Performance
> > > | Model          | Elliptic | Tolokers | YelpChi |
> > > |----------------|----------|----------|---------|
> > > | GADM (random)  | 95.76    | 77.79    | 85.28   |
> > > | CGADM (ours)   | 99.30    | 79.68    | 92.69   |
> > >
> > > From the results, it is evident that integrating the prior into both the forward and reverse diffusion chains significantly enhances anomaly detection performance. This approach effectively constructs a denoising diffusion probabilistic model tailored for anomaly detection, highlighting the value of leveraging prior information.
> > >
> > > ### Enhanced Visualization for Figure 1
> > > We also acknowledge your and Reviewer Bfk9's feedback regarding the intuitiveness of Figure 1. To improve clarity and informativeness, we replaced the synthetic distributions in Figure 1 with real data distributions. Specifically, we sampled 20 anomalous nodes and 20 normal nodes from the *Elliptic* dataset and visualized the following:
> > >
> > > 1. The ground truth distributions of anomalous and normal nodes.
> > > 2. Distributions under two initialization scenarios: **random initialization** and **A-priori initialization**.
> > > 3. The final distributions after the diffusion process, demonstrating the impact of the prior-guided diffusion mechanism.
> > >
> > > These modifications ensure that the visualization is more convincing and better illustrates how prior information enhances the generation of a meaningful joint distribution, ultimately improving anomaly detection performance.
> > >
> > > We hope these additions address your concerns and further clarify the contributions of our proposed approach. Thank you again for your valuable feedback, which has significantly improved the quality and rigor of our work.

---

> > > ### Author Response · Authors · 2024-11-24
> > > **Further Response to Reviewer MkJ2 (Part 2)**
> > >
> > > >**Q3**: At the feature level, the authors introduce a detailed claim that the denoising reconstruction was proposed to address the challenge of malicious feature manipulation. In addition to the camouflage of anomalies, the contamination of normal data also affects performance. I agree that CGADM has advantages over discriminative models, and I suggest the authors explore further evidence regarding the role of denoising reconstruction.
> > >
> > >
> > > **A3**:
> > >
> > > We greatly appreciate the reviewer’s insightful feedback regarding the role of denoising reconstruction in addressing malicious feature manipulation. We fully agree that further evidence is necessary to demonstrate this aspect, and we have conducted additional experiments to explore how CGADM handles feature-level perturbations.
> > >
> > > ### Experimental Setup
> > > To evaluate the robustness of CGADM against feature manipulation, we introduced feature perturbations in the *Elliptic* and *Tolokers* datasets. Specifically, we randomly perturbed the features of nodes with varying proportions (10%, 20%, and 30%) by randomly selecting values from their possible ranges with uniform probability. We then compared the performance of CGADM with GHRN (the best-performing baseline from our original experiments) under these conditions.
> > >
> > > The results are summarized below:
> > >
> > > #### AUPRC Performance
> > > | Perturbation Rate | Dataset | CGADM   | GHRN    | CGADM Change | GHRN Change |
> > > |-------------------|---------|---------|---------|--------------|-------------|
> > > | 0%                | Elliptic| 97.03   | 88.13   | —            | —           |
> > > | 10%               | Elliptic| 87.61   | 78.77   | -9.70%       | -10.62%     |
> > > | 20%               | Elliptic| 85.64   | 71.11   | -11.74%      | -19.31%     |
> > > | 30%               | Elliptic| 78.07   | 64.72   | -19.54%      | -26.56%     |
> > > | 0%                | Tolokers| 46.02   | 45.25   | —            | —           |
> > > | 10%               | Tolokers| 40.21   | 38.87   | -12.62%      | -14.90%     |
> > > | 20%               | Tolokers| 37.55   | 34.90   | -18.40%      | -22.87%     |
> > > | 30%               | Tolokers| 34.51   | 30.45   | -25.00%      | -32.70%     |
> > >
> > > #### AUROC Performance
> > > | Perturbation Rate | Dataset | CGADM   | GHRN    | CGADM Change | GHRN Change |
> > > |-------------------|---------|---------|---------|--------------|-------------|
> > > | 0%                | Elliptic| 99.30   | 97.04   | —            | —           |
> > > | 10%               | Elliptic| 91.68   | 88.75   | -7.67%       | -8.55%      |
> > > | 20%               | Elliptic| 91.14   | 84.31   | -8.22%       | -13.12%     |
> > > | 30%               | Elliptic| 85.65   | 76.79   | -13.74%      | -20.87%     |
> > > | 0%                | Tolokers| 79.68   | 77.98   | —            | —           |
> > > | 10%               | Tolokers| 71.48   | 69.61   | -10.29%      | -10.74%     |
> > > | 20%               | Tolokers| 68.59   | 64.41   | -13.92%      | -17.40%     |
> > > | 30%               | Tolokers| 63.69   | 60.18   | -20.06%      | -22.83%     |
> > >
> > > ### Observations
> > > As the proportion of perturbed nodes increases, the performance of both models decreases. However, CGADM consistently exhibits a slower decline compared to GHRN. This highlights CGADM’s superior robustness to feature perturbations, which we attribute to its denoising reconstruction mechanism. This mechanism leverages information from neighboring nodes during the reverse diffusion process to iteratively restore the true anomaly signals.
> > >
> > >
> > > ### Visualization
> > > To further illustrate this robustness, we included a set of performance curves in the appendix. These curves clearly show how CGADM maintains stronger resilience as the proportion of perturbed nodes increases.
> > >
> > > We hope these additional experiments and visualizations address your concerns and provide a clearer understanding of the advantages of denoising reconstruction in CGADM. Thank you again for your valuable feedback, which has significantly enhanced the rigor and depth of our study.

---

> > > ### Author Response · Authors · 2024-11-24
> > > **Further Response to Reviewer MkJ2 (Part 3)**
> > >
> > > >**Q4**: Similarly, at the topology level, the authors claim that adding GNN layers improves performance, demonstrating that over-smoothing is effectively addressed. However, are there any metrics or experiments that could demonstrate the high-frequency components indeed reflected in the residual propagations? The explanation based solely on Figure 2 is insufficient to fully support this claim.
> > >
> > >
> > >
> > >
> > > **A4**:
> > > We sincerely thank the reviewer for their insightful comment. We agree that the explanation in Figure 2 alone may not sufficiently support this claim, and we have conducted additional experiments to provide clearer evidence.
> > >
> > > ### Theoretical Insight
> > > The high-frequency components in our approach stem from the operation:
> > >
> > > $$
> > > \mathbf{h_v^{l-1}} - \frac{1}{|\mathcal{N}(v)|} \sum_{u \in \mathcal{N}(v)} \mathbf{h_u^{l-1}}.
> > > $$
> > >
> > > As explained in GCNII (Chen et al., 2020), the aggregation operation in graph convolution averages the features of neighboring nodes, acting as a low-pass filter that preserves low-frequency signals. By subtracting this low-frequency signal from the original features, the residual propagation effectively retains the high-frequency information. This high-frequency information is particularly critical in anomaly detection, as it allows the model to distinguish anomalous nodes from normal nodes more effectively.
> > >
> > > ### Ablation Study
> > > To further substantiate that the high-frequency components are indeed reflected in the residual propagations, we designed an ablation study comparing our original CGADM (denoted as $CGADM_{HP}$) with a variant (denoted as $CGADM_{LP}$) that only propagates low-frequency signals. In $CGADM_{LP}$, the graph convolution operation is replaced with the standard GCN:
> > >
> > > $$
> > > \frac{1}{|\mathcal{N}(v)| + 1} \left( \mathbf{h_v^{l-1}} + \sum_{u \in \mathcal{N}(v)} \mathbf{h_u^{l-1}} \right),
> > > $$
> > >
> > > where the feature representation is averaged across the node and its neighbors, propagating only low-frequency signals.
> > >
> > > We conducted experiments on the *Elliptic* and *YelpChi* datasets, varying the number of GNN layers in the denoiser module. The results are as follows:
> > >
> > > #### Performance Comparison
> > > | GNN Layers | Model          | AUPRC (Elliptic) | AUROC (Elliptic) | AUPRC (YelpChi) | AUROC (YelpChi) |
> > > |------------|----------------|------------------|------------------|-----------------|-----------------|
> > > | 1          | $CGADM_{HP}$ | 97.13            | 99.22            | 75.04           | 92.37           |
> > > |            | $CGADM_{LP}$ | 95.71            | 98.43            | 72.23           | 91.88           |
> > > | 2          | $CGADM_{HP}$ | 97.31            | 99.38            | 75.20           | 92.62           |
> > > |            | $CGADM_{LP}$ | 93.73            | 97.60            | 70.92           | 90.88           |
> > > | 3          | $CGADM_{HP}$ | 97.32            | 99.44            | 76.54           | 92.69           |
> > > |            | $CGADM_{LP}$ | 90.83            | 95.58            | 71.43           | 89.64           |
> > > | 4          | $CGADM_{HP}$ | 97.53            | 99.44            | 77.27           | 93.05           |
> > > |            | $CGADM_{LP}$ | 87.12            | 92.60            | 69.98           | 87.71           |
> > > | 5          | $CGADM_{HP}$ | 97.57            | 99.50            | 77.29           | 92.92           |
> > > |            | $CGADM_{LP}$ | 81.20            | 89.49            | 68.71           | 86.08           |
> > >
> > > ### Observations
> > > 1. **High-Frequency Signal Preservation Matters**:
> > >    $CGADM_{HP}$, which retains high-frequency signals through residual propagation, consistently outperforms $CGADM_{LP}$ across all metrics and datasets. This highlights the importance of preserving high-frequency information for anomaly detection, as anomalies often manifest as local deviations that are captured by these components.
> > >
> > > 2. **Sensitivity to GNN Layers**:
> > >    For $CGADM_{LP}$, performance declines significantly as the number of GNN layers increases. This is indicative of the well-known over-smoothing issue, where stacking multiple low-pass filters causes node representations to converge, losing discriminative information. Conversely, $CGADM_{HP}$ remains robust, and its performance even improves slightly with additional layers, demonstrating the effectiveness of residual propagation in mitigating over-smoothing.
> > >
> > > 3. **Iterative Refinement Amplifies Over-Smoothing**:
> > >    In the context of our diffusion model, the iterative refinement process repeatedly aggregates neighborhood information, exacerbating the impact of over-smoothing in $CGADM_{LP}$. This leads to a failure to capture new anomaly-relevant signals at each stage of refinement. In contrast, $CGADM_{HP}$ avoids this issue by leveraging high-frequency signals to refine anomaly detection throughout the iterative process.
> > >
> > >
> > > We hope this additional evidence and analysis address your concerns. Thank you again for your valuable feedback, which has helped us strengthen our contributions and theoretical explanations.

---

> > > > ### Comment · Reviewer_MkJ2 · 2024-11-24
> > > >
> > > > Dear authors
> > > >
> > > > I sincerely appreciate the newly added clarifications and experiments, which have addressed part of my concerns. I agree that the proposed conditional diffusion framework without reliance on augmentation strategies could be an effective approach for GAD, as supported by the empirical evidence provided in the rebuttal.   At the same time, I feel that the current version of the introduction is somewhat disorganized. If the challenges are presented from the perspective of feature and topology, I think the second point about improving efficiency through sampling does not align well with this challenge. The authors need to carefully consider how to structure the storyline for this aspect.  The current version of Figure 1 still requires additional information to help readers understand it better. It does not illustrate the internal working principles but only presents the inputs and outputs.
> > > > Overall,  I would like to raise my score to 6; however, I feel that the current draft may require substantial revisions, especially given the extensive experiments are added during the rebuttal phase. Consequently, I have lowered my confidence to 3. I hope my feedback can help you further refine and strengthen the paper and good luck.

---

> > > > > ### Author Response · Authors · 2024-11-26
> > > > > **Additional Response to Reviewer MkJ2**
> > > > >
> > > > > Thank you for your thoughtful feedback and for raising your score. We appreciate your recognition of the strengths of our work and the detailed suggestions for further improvement.
> > > > >
> > > > > ---
> > > > >
> > > > > ### Clarification on the Storyline
> > > > >
> > > > > We acknowledge your concerns regarding the coherence of the storyline in our introduction. To clarify:
> > > > >    - We initially highlight two primary challenges in graph anomaly detection: **feature-level challenges** and **topology-level challenges** .
> > > > >    - We propose diffusion model can addresses these challenges through its **denoising construction** and **iterative refinement**.
> > > > >    - While diffusion models show promise for addressing the above challenges, their application to graph anomaly detection introduces new **effectiveness** and **efficiency** challenges.
> > > > >    - We tackle these with two novel components:
> > > > >      - A **prior-guided diffusion process** that improves anomaly detection effectiveness by incorporating domain knowledge into the diffusion process.
> > > > >      - A **prior confidence-aware mechanism** that dynamically adjusts reverse sampling steps, significantly enhancing computational efficiency on large-scale graphs.
> > > > >
> > > > >
> > > > > ### Addressing Revisions During the Rebuttal Phase
> > > > >
> > > > > We appreciate your acknowledgment of the extensive experiments and additions made during the rebuttal phase. These contributions were inspired by the productive dialogue with reviewers and aimed to strengthen the empirical evidence for our method. Moving forward, we will:
> > > > > - Reorganize the added content to integrate it seamlessly into the paper, ensuring a logical flow and consistency with the overall storyline.
> > > > > - Refine the manuscript further to improve clarity and structure, reducing redundancy introduced during the rapid revisions.
> > > > >
> > > > >
> > > > > Thank you again for your time and valuable insights.
> > > > >
> > > > > Sincerely,
> > > > > Authors

---

### Official Review · Reviewer_Jvy3 · 2024-10-30

**Soundness:** 3
**Presentation:** 2
**Contribution:** 2
**Rating:** 5
**Confidence:** 4

**Summary:**

This paper proposes a Conditional Graph Anomaly Diffusion Model (CGADM) for generative graph anomaly detection, aiming at considering the interdependencies of node anomalies from a holistic graph perspective.

**Strengths:**

1. The performance of CGADM outperforms the SOTA methods on most of the datasets on most of the metrics.

2. Malicious nodes in a dynamic adversarial environment are very hard to detect since they are normal in the moment but can be anomalous in the next second. It is a worth studying problem, and pattern recognition may be a plausible approach.

3. The method is efficient due to the reduction of the reverse steps, time and space scalability is always important in graph algorithms.

**Weaknesses:**

1. The performance gap on Quest and Reddit are somehow marginal. Diffusion model need large amount of data to capture the distribution, and according to Table 3, Quest and Reddit have the least anomaly ratio. This raises the question that: In real world, anomaly ratio may be less than 1%, will CGADM still work in imbalanced scenarios?

2. The distribution definition is not clear. What indeed is the distribution that the authors expect DM to capture? TSNE visualizations or some insights about this are appreciated. Figure 1 doesn't provide enough information in such an important place.

3. Some of the related work [1~4] (especially new works in the field) are missing.

[1] Gao et al. Alleviating Structural Distribution Shift in Graph Anomaly Detection.

[2] Xu et al. Revisiting graph-based fraud detection in sight of heterophily and spectrum.

[3] Qiao et al. Generative Semi-supervised Graph Anomaly Detection.

[4] He et al. ADA-GAD: Anomaly-Denoised Autoencoders for Graph Anomaly Detection.

**Questions:**

Will CGADM still work in imbalanced scenarios?

---

> ### Author Response · Authors · 2024-11-21
> **Response to Reviewer Jvy3**
>
> Thanks for your detailed reading and suggestions. Please see our detailed response and clarification below:
>
> >**Q1**: The performance gap on Quest and Reddit are somehow marginal. Diffusion model need large amount of data to capture the distribution, and according to Table 3, Quest and Reddit have the least anomaly ratio. This raises the question that: In real world, anomaly ratio may be less than 1%, will CGADM still work in imbalanced scenarios?
>
> **A1**: Thank you for your thoughtful feedback and for raising the concern regarding the marginal performance gap observed on the Quest and Reddit datasets, as well as the question about CGADM’s efficacy in highly imbalanced scenarios. We appreciate the opportunity to clarify these points.
>
> 1. **Performance Gap on Quest and Reddit**
>    As observed in Table 3, the anomaly ratio for Quest and Reddit datasets is indeed relatively low, which presents challenges typical of highly imbalanced datasets. However, it is important to interpret the performance using **both AUROC and AUPRC** metrics. While AUROC provides a comprehensive view of the model’s discriminative power, it is often less sensitive to the performance on the minority (anomalous) class in imbalanced scenarios. Conversely, AUPRC focuses exclusively on the precision and recall of the positive (anomalous) class, making it more indicative of a model's true capability in anomaly detection.
>
>    On the Quest dataset, our method achieves an **AUPRC of 18.51**, significantly surpassing the previous SOTA method BWGNN (14.64), with an improvement exceeding **26%**. This result underscores CGADM's ability to achieve both high precision and recall in detecting anomalies, which is critical in real-world imbalanced scenarios.
>    On the Reddit dataset, while our AUPRC is slightly behind GHRN by **1%**, our method achieves the highest AUROC. Given the inherent variability in experimental results, we consider this difference within the margin of error, suggesting that our method remains competitive on this dataset.
>
>    **Generative Paradigm as Complementary**:  CGADM introduces a novel generative framework for anomaly detection, which is orthogonal to the discriminative improvements of existing methods. Importantly, the flexibility of CGADM allows the integration of more advanced prior estimators, which could further enhance its performance when combined with state-of-the-art discriminative techniques.
>
>    When evaluated holistically across five datasets, CGADM consistently demonstrates superior performance compared to existing methods, highlighting its generalizability and robustness.
>
> 2. **Efficacy in Highly Imbalanced Scenarios**
>    To directly address the reviewer’s question—“Will CGADM still work in imbalanced scenarios?”—we conducted additional experiments on the **DGraph** dataset [1], a highly imbalanced real-world financial fraud detection dataset where anomalies constitute only **1.3%** of the data. The results are presented below:
>
>     | **Method**   | **AUPRC** | **AUROC** |
>     |--------------|-----------|-----------|
>     | GCN          | 3.66      | 74.97     |
>     | GIN          | 3.22      | 73.14     |
>     | GraphSAGE    | 3.43      | 73.81     |
>     | GAT          | 3.65      | 75.17     |
>     | GAS          | 2.91      | 71.21     |
>     | PCGNN        | 2.82      | 71.78     |
>     | BWGNN        | 3.63      | 75.16     |
>     | GHRN         | 3.68      | 75.15     |
>     | **CGADM**    | **3.83**  | **76.43** |
>
>    As the tables illustrate, CGADM consistently outperforms all baseline methods on both AUPRC and AUROC metrics in this **extremely imbalanced setting**. Notably, the AUPRC metric demonstrates CGADM's ability to handle rare event detection by excelling in anomaly-specific precision and recall. Similarly, the superior AUROC indicates robust overall discriminative performance.
>
> 3. **Discussion on Data Imbalance and Diffusion Models**
>    The reviewer correctly noted that diffusion models often require large datasets to effectively capture distributions. In CGADM, we mitigate this limitation through the **prior-guided denoising diffusion probability model**, which enables the framework to leverage prior knowledge about the anomaly distribution, thus reducing reliance on extensive data samples.
>
>
> Thank you again for your valuable feedback.
>
> **References**
> [1] Huang X, Yang Y, Wang Y, et al. Dgraph: A large-scale financial dataset for graph anomaly detection[J]. Advances in Neural Information Processing Systems, 2022, 35: 22765-22777.

---

> ### Author Response · Authors · 2024-11-21
> **Response to Reviewer Jvy3 (Part 2)**
>
> >**Q2**: The distribution definition is not clear. What indeed is the distribution that the authors expect DM to capture? TSNE visualizations or some insights about this are appreciated. Figure 1 doesn't provide enough information in such an important place.
>
>
> **A2**:
>
> Thank you for your insightful feedback regarding the clarity of the distribution definition and the need for additional visualization to better convey the distribution captured by our model. We have carefully considered your comments and revised Figure 1 to provide a more informative and intuitive representation of the joint anomaly distribution and the role of our model in capturing it. Please refer to the revised manuscript for the updated figure and accompanying explanation.
>
>
>
> ---
>
>
>
>
>
>
> >**Q3**: Some of the related work [1~4] (especially new works in the field) are missing
>
>
> **A3**:
>
> We greatly appreciate your comments and suggestions, particularly regarding the inclusion of relevant recent work in the related work section of our manuscript. Below, we provide a comprehensive discussion of the works you mentioned in the revision.
>
> Recent advancements in graph anomaly detection have tackled various challenges. Gao et al. [1] addressed structural distribution shifts through feature-specific constraints in Graph Decomposition Networks (GDN), while Xu et al. [2]proposed SEC-GFD to handle heterophily and label imbalance via spectral filtering. Qiao et al. [3] introduced a semi-supervised generative framework (GGAD) that leverages labeled normal nodes to generate pseudo-anomalies, and He et al. [4] developed ADA-GAD to mitigate anomaly overfitting through anomaly-denoised graph augmentation. Unlike these methods, our CGADM adopts a novel generative diffusion approach to model the joint anomaly distribution over the graph, enabling holistic and scalable anomaly detection without reliance on labeled data or augmentation strategies.
>
>
> We hope this response provides the necessary context and clarity regarding the related work and our approach. We have made the appropriate revisions to the manuscript and are happy to further discuss any additional questions you may have.
>
> Thank you again for your valuable feedback.
>
>
> **References**
> [1] Gao et al. Alleviating Structural Distribution Shift in Graph Anomaly Detection.
> [2] Xu et al. Revisiting graph-based fraud detection in sight of heterophily and spectrum.
> [3] Qiao et al. Generative Semi-supervised Graph Anomaly Detection.
> [4] He et al. ADA-GAD: Anomaly-Denoised Autoencoders for Graph Anomaly Detection.

---

> > ### Comment · Reviewer_Jvy3 · 2024-11-23
> >
> > Dear author,
> >
> > Thanks for the rebuttal. The experiment on DGraph is promising. However, the "distribution" in Figure 1 is still too intuitive. I believe plotting the distribution from the real data might be helpful.

---

> > > ### Author Response · Authors · 2024-11-24
> > > **Further Response to Reviewer Jvy3**
> > >
> > > Thank you for your insightful feedback and appreciation of our experimental results on the DGraph dataset. We value your suggestion regarding the intuitiveness of Figure 1 and have revised it based on your input.
> > >
> > > To address your concern, we replaced the synthetic distribution in Figure 1 with real data distributions to provide a more informative and convincing visualization. Specifically, we sampled 20 anomalous nodes and 20 normal nodes from the _Elliptic_ dataset and plotted their ground truth distributions alongside the distributions obtained under two initialization scenarios: **random initialization** and **A-priori initialization**. Additionally, we included the final distributions after our diffusion process for better clarity.
> > >
> > > This updated figure demonstrates the necessity and effectiveness of our **prior-guided diffusion process**, which integrates a pre-trained conditional anomaly estimator into both the forward and reverse diffusion chains.
> > >
> > > We believe these additions not only make Figure 1 more illustrative but also solidify the validity of our methodology.
> > >
> > > Thank you once again for your constructive comments, which have greatly contributed to improving the clarity and rigor of our work. We hope the revised Figure 1 addresses your concerns effectively.

---

> ### Author Response · Authors · 2024-11-29
>
> Dear Reviewer,
>
> I sincerely thank you for your valuable feedback during the review process. Following your insightful suggestions, we have made substantial revisions to our paper. It is gratifying to hear that you are satisfied with the additional experiments we conducted. Furthermore, in response to your recommendation regarding plotting the distribution from the real data, we have addressed this in the revised version, which we kindly invite you to review **[Further Response to Reviewer Jvy3](https://openreview.net/forum?id=saRBktzh3q&noteId=HYbToxD6HG)** in detail.
>
> We truly appreciate the conference organizers for extending the discussion period by six days, allowing for more thorough engagement. As the deadline for discussions approaches, we would be immensely grateful if you could review our latest responses and consider revisiting your scores based on the revisions and updates we’ve provided.
>
> Your constructive feedback has been instrumental in enhancing the quality of our work, and we deeply value your contribution to this process. If you have any further questions or require additional clarifications, please do not hesitate to reach out. I am more than happy to provide any necessary explanations.
>
> Thank you again for your support and thoughtful review.
>
> Best regards,
> Authors

---

### Author Response · Authors · 2024-11-22
**Revised PDF has been uploaded**

Dear reviewers, I sincerely apologize for the delay in uploading the revised PDF. The updated version has now been uploaded. If you have any new questions, I warmly welcome you to discuss them with me in the comments.

---

### Note · Authors · 2025-01-29

I have read and agree with the venue's withdrawal policy on behalf of myself and my co-authors.